# Conformal Prediction with Cellwise Outliers: A Detect-then-Impute Approach

**Qian Peng** [1]  **Yajie Bao** [1]  **Haojie Ren** [2]  **Zhaojun Wang** [1]  **Changliang Zou** [1]

## Abstract

Conformal prediction is a powerful tool for constructing prediction intervals for black-box models, providing a finite sample coverage guarantee for exchangeable data. However, this exchangeability is compromised when some entries of the test feature are contaminated, such as in the case of cellwise outliers. To address this issue, this paper introduces a novel framework called *detect-then-impute conformal prediction*. This framework first employs an outlier detection procedure on the test feature and then utilizes an imputation method to fill in those cells identified as outliers. To quantify the uncertainty in the processed test feature, we adaptively apply the detection and imputation procedures to the calibration set, thereby constructing exchangeable features for the conformal prediction interval of the test label. We develop two practical algorithms, `PDI-CP` and `JDI-CP`, and provide a distribution-free coverage analysis under some commonly used detection and imputation procedures. Notably, `JDI-CP` achieves a finite sample $1 - 2\alpha$ coverage guarantee. Numerical experiments on both synthetic and real datasets demonstrate that our proposed algorithms exhibit robust coverage properties and comparable efficiency to the oracle baseline.

## 1. Introduction

As the volume and complexity of data expand, machine learning models and algorithms have become essential tools for enhancing decision-making processes across various domains, including autonomous driving (Min et al., 2024), wing design (Zhang et al., 2024), and disease diagnosis (Shang et al., 2024). Ensuring the reliability of

---

[1]School of Statistics and Data Sciences, LPMC, KLMDASR and LEBPS, Nankai University, Tianjin, China [2]School of Mathematical Sciences, Shanghai Jiao Tong University, Shanghai, China. Correspondence to: Yajie Bao <yajiebao@nankai.edu.cn>, Haojie Ren <haojieren@sjtu.edu.cn>.

*Proceedings of the $42^{nd}$ International Conference on Machine Learning*, Vancouver, Canada. PMLR 267, 2025. Copyright 2025 by the author(s).

prediction models in risk-sensitive applications hinges on valid uncertainty quantification. Conformal prediction (CP, Vovk et al. (2005)) offers a flexible and robust framework for uncertainty quantification of arbitrary machine learning models by constructing prediction interval (PI). To be specific, suppose we have collected an i.i.d. labeled dataset $\mathcal{D}_l = \{(X_i, Y_i)\}_{i=1}^n \subset \mathbb{R}^d \times \mathbb{R}$ where $(X_i, Y_i) \sim P = P_X \times P_{Y|X}$. Given the test data $X_{n+1}$, CP issues a PI $\widehat{C}^{\text{CP}}(X_{n+1})$ satisfying the marginal coverage $\mathbb{P}\{Y_{n+1} \in \widehat{C}^{\text{CP}}(X_{n+1})\} \geq 1 - \alpha$ if the test data and labeled data are exchangeable. However, the distribution of some variables in test data may deviate from that of the training data due to measurement error, natural variation, or feature contamination. This issue, referred to as *cellwise outliers* (Alqallaf et al., 2009), can result in inaccurate predictions and flawed decisions. For example, consider a hospital that seeks to use the patient's biomarkers to predict incidence rate based on a model trained on data from recovered patients. For a newly admitted patient, some of whose biomarkers may differ from those in the training set. Conducting robust uncertainty quantification for machine learning models in the presence of cellwise outliers remains an underexplored area.

In this case, we observe a contaminated test feature $\tilde{X}_{n+1}$ with entries

$$\tilde{X}_{n+1,j} = \begin{cases} X_{n+1,j}, & j \in [d] \setminus \mathcal{O}^* \\ Z_{n+1,j}, & j \in \mathcal{O}^* \end{cases}, \qquad (1)$$

where $\mathcal{O}^* \subseteq [d]$ denotes coordinates of cellwise outliers in $\tilde{X}_{n+1}$, and $Z_{n+1} \sim P_Z$ is an arbitrarily distributed outliers and independent of $\{(X_i, Y_i)\}_{i=1}^{n+1}$. Our goal is to build a PI for test label $Y_{n+1}$ with a target coverage level $1 - \alpha$ when only $\tilde{X}_{n+1}$ is observed. However, the classical CP method is incapable of providing a valid PI as test data $(\tilde{X}_{n+1}, Y_{n+1})$ is not exchangeable with the labeled data $\mathcal{D}_l$. Furthermore, the distribution shift from the labeled data to the test data is unknown and difficult to estimate without additional distribution or structure assumptions, which makes the weighted conformal prediction (WCP) method (Tibshirani et al., 2019) that requires consistent likelihood ratio estimation unsuitable in this context as well. Therefore, a more principled CP approach is needed to address cellwise outliers in test data.

To deal with cellwise outliers, our framework starts from the common approach, *detect then impute* (Raymaekers & Rousseeuw, 2021; 2024b). We implement a detection method to identify the locations of outliers in $\tilde{X}_{n+1}$ and mask those entries, followed by using an imputation method to impute the masked entries. After this procedure, we obtain the processed feature $\check{X}_{n+1}^{\mathrm{DI}}$ (see Section 3.1 for details). The next step is to construct the PI using the labeled data $\mathcal{D}_l$ and $\check{X}_{n+1}^{\mathrm{DI}}$. However, $\check{X}_{n+1}^{\mathrm{DI}}$ and $X_i$ for $i \in [n]$ remain nonexchangeable because the detection and imputation procedures cannot exactly recover the uncontaminated feature $X_{n+1}$. To tackle this challenge, our key strategy involves adaptively applying detection and imputation procedures to the labeled features $X_i$ for $i \in [n]$, aiming to produce exchangeable copies of the processed test feature $\check{X}_{n+1}^{\mathrm{DI}}$. Subsequently, we utilize the split conformal prediction (SCP) method (Papadopoulos et al., 2002; Vovk et al., 2005) to construct the PI based on the residuals computed from the processed labeled data. Building on this concept, we first propose an oracle detection-imputation conformal prediction (ODI-CP) method and prove that it can achieve a finite sample coverage property. Following this, we develop a *proxy detection-imputation* CP (PDI-CP) method based on the data-driven approximation to ODI-CP. To further have a finite sample coverage guarantee, we propose another algorithm *joint detection-imputation* CP (JDI-CP) to construct Jackknife+ type PI (Barber et al., 2021).

Our contributions have three folds: (1) we propose a novel CP framework to efficiently address the issue of cellwise outliers in test data, which can be wrapped around any well-known detection and imputation procedure; (2) for arbitrarily cellwise outliers, we establish coverage error bounds for PDI-CP and a finite sample $1 - 2\alpha$ coverage property for JDI-CP, and all results are distribution-free; (3) experiments on synthetic data and real data show that the proposed method has a robust coverage control and a comparable performance with oracle approaches.

## 2. Related work

**Cellwise outliers.** In the context of handling extensive datasets, it is increasingly common for only a few individual entries (cells) to show anomalies, called cellwise outliers, first published by Alqallaf et al. (2009). Building upon this concept, Van Aelst et al. (2011) proposed the earliest cellwise detection method and Rousseeuw & Bossche (2018) pioneered the Detect Deviating Cells (DDC) algorithm, which predicts the value for each cell and identifies outliers by the significant deviations of predicted values from the original values. Thereafter, Raymaekers & Rousseeuw (2021) proposed the cellHandler method, which integrates all relations in variables to identify complex adversarial cellwise outliers. Along this line, Zaccaria et al. (2024) proposed the cell-

GMM method to detect cellwise outliers in heterogeneous populations. Inspired by these detection methods, Liu et al. (2022) proposed the BiRD algorithm to control the false discovery rate in detection. Besides detection, there are various cellwise robust methods available for cases including discriminant analysis (Aerts & Wilms, 2017), principal component analysis (Hubert et al., 2019), regression models (Öllerer et al., 2016; Filzmoser et al., 2020), cluster analysis (García-Escudero et al., 2021), location and covariance estimators (Raymaekers & Rousseeuw, 2024a) and so on. To the best of our knowledge, this paper is the first work considering the predictive inference problem with cellwise outliers.

**Conformal prediction without exchangeability.** Our paper is related to the line of work on the CP approaches catered to nonexchangeable data. Chernozhukov et al. (2018) extended CP to time series data based on block-permutation. Tibshirani et al. (2019) proposed the weighted conformal prediction (WCP) framework to deal with the covariate shift between labeled data and test data. Further, Yang et al. (2024) developed a doubly robust approach to construct PIs satisfying approximate marginal coverage under covariate shift. Podkopaev & Ramdas (2021) studied the conformal PI under the label shift case. For arbitrary distribution shift issues, Barber et al. (2023) developed a robust weighted CP method with deterministic weights that could provide approximately marginal coverage under gradual changes in the data distribution. Cauchois et al. (2024) studied constructing distributional robust PI for the test data sampled from a distribution in an $f$-divergence ball around the population of labeled data. This paper focuses on discussing another special case of nonexchangeability caused by cellwise outliers, which cannot be addressed by the methods proposed so far.

**Predictive inference with missing data.** Another topic closely related to our paper is the construction of PI in the presence of missing values in features. The entries identified as cellwise outliers by the detection procedure are masked, transforming the primary problem into predictive inference with missing data. Zaffran et al. (2023; 2024) examined a setting where missing values may occur in both labeled and test features, proposing to perform SCP after imputing the missing entries in both sets. This approach could achieve finite-sample marginal coverage if the random missing patterns in the labeled and test sets are exchangeable. However, this assumption does not hold in our context, as the masked entries identified by the detection procedure lack exchangeability due to cellwise outliers in the test feature. In addition, Lee et al. (2024) proposed the propensity score discretized conformal prediction (pro-CP) to construct PIs with missing values in outcomes. This is different from the scenario considered in this paper.

**Conformal inference for outlier detection.** In recent years, conformal inference has emerged as an important tool in the field of outlier detection at the intersection of statistics and machine learning. Due to the space limit, we list some relevant works (Guan & Tibshirani, 2022; Zhang et al., 2022; Bates et al., 2023b; Marandon et al., 2024; Bashari et al., 2024; Liang et al., 2024). However, the above works aim to apply conformal inference to test the presence of outliers, while this paper focuses on the uncertainty quantification of prediction and the construction of PI when some entries of the observed test feature are contaminated.

# 3. Problem setup and oracle approach

In this section, we present problem formulation and motivation of our method from an oracle perspective. Before that, we list some useful notations. We split the labeled data $\mathcal{D}_l$ into the training set $\mathcal{D}_t = \{(X_i, Y_i)\}_{i=1}^{n_0-1}$ and the calibration set $\mathcal{D}_c = \{(X_i, Y_i)\}_{i=n_0}^{n}$, where $1 < n_0 < n$. The prediction model $\hat{\mu}(\cdot) : \mathbb{R}^d \to \mathbb{R}$ is fitted on the training set. Given real numbers $\{r_i\}_{i=1}^{n}$ and $\alpha \in (0,1)$, we denote $\hat{q}_\alpha^-(\{r_i\}_{i=1}^{n})$ and $\hat{q}_\alpha^+(\{r_i\}_{i=1}^{n})$ as the $\lceil \alpha(n+1) \rceil$th and $\lceil (1-\alpha)(n+1) \rceil$th smallest value in $\{r_i\}_{i=1}^{n}$, respectively. Given $x \in \mathbb{R}^d$, we write the $\ell$-th coordinate of $x$ as $x_\ell$.

## 3.1. Detection and imputation procedures

This subsection introduces the detection ($\mathbf{D}$) and imputation ($\mathbf{I}$) procedures to clean the test feature for constructing PI.

A generic detection procedure ($\mathbf{D}$) includes two ingredients: cellwise score function $\hat{s}_j(\cdot) : \mathbb{R} \to \mathbb{R}$ and threshold $\tau_j \in \mathbb{R}$ for $j \in [d]$, which are both fitted on the training set $\mathcal{D}_t$. Given any $x \in \mathbb{R}^d$, we write the output of detection procedure as $\mathbf{D}(x) = \{j \in [d] : \hat{s}_j(x_j) > \tau_j\}$. For example, Z-score (Curtis et al., 2016) is a detection method based on standardization, which has score $z_j := \hat{s}_j(x_j) = (x_j - \mu_j)/\sigma_j$, where $\mu_j$ and $\sigma_j$ are the mean and standard deviation of the j-th coordinate. Typically, the entry $x_j$ is considered as an outlier when $|z_j| > 3$. Other more sophisticated detection methods include DDC (Rousseeuw & Bossche, 2018), one-class SVM classifier (Bates et al., 2023a), etc.

We define the imputation procedure ($\mathbf{I}$) as a function of feature $x \in \mathbb{R}^d$ and entry subset $\mathcal{O} \subseteq [d]$, then the imputed value for coordinate $j \in \mathcal{O}$ is given by

$$[\mathbf{I}(x, \mathcal{O})]_j = \begin{cases} x_j, & j \notin \mathcal{O} \\ \phi_j(\{x_l\}_{l \notin \mathcal{O}}), & j \in \mathcal{O} \end{cases},$$

where $\phi_j(\cdot) : \mathbb{R}^{d_{\mathrm{obs}}} \to \mathbb{R}$ with $d_{\mathrm{obs}} = d - |\mathcal{O}|$ is imputation function for the $j$-th coordinate, and is also fitted on the training set $\mathcal{D}_t$. The output of most popular imputation methods can be formalized in the above form, such as Mean Imputation, k-Nearest Neighbour (kNN, Troyanskaya et al.

(2001)) and Multivariate Imputation by Chained Equations (MICE, Van Buuren et al. (1999)).

Hereafter, we abbreviate the detection and imputation procedure as $\mathbf{DI}$, and denote the processed test feature as

$$\check{X}_{n+1}^{\mathrm{DI}} = \mathbf{I}(\tilde{X}_{n+1}, \tilde{\mathcal{O}}_{n+1}), \ \tilde{\mathcal{O}}_{n+1} = \mathbf{D}(\tilde{X}_{n+1}). \quad (2)$$

## 3.2. Oracle detection-imputation

Because $\mathbf{DI}$ cannot exactly recover the ground truth feature $X_{n+1}$, i.e., $\check{X}_{n+1}^{\mathrm{DI}} \neq X_{n+1}$. Therefore, we must consider the recovery uncertainty from $\mathbf{DI}$ when constructing the PI. We begin with an oracle procedure by assuming $\mathcal{O}^*$ is known and $\mathbf{D}$ satisfies the following sure detection assumption. Later we will discuss the necessity of this assumption for informative coverage in Section 3.3.

**Assumption 3.1** (Sure detection). We call the detection procedure $\mathbf{D}$ has the sure detection property, if all the cellwise outliers are detected, namely $\mathcal{O}^* \subseteq \tilde{\mathcal{O}}_{n+1} = \mathbf{D}(\tilde{X}_{n+1})$ for test point $\tilde{X}_{n+1}$.

**Assumption 3.2** (Isolated detection). Given the score function $\hat{s}_j$ and threshold $\tau_j$, the detection indicator $\mathbb{1}\{\hat{s}_j(x_j) \leq \tau_j\}$ depends only on $x_j$ for $j \in [d]$.

Denote $\hat{\mathcal{O}}_{n+1} = \mathbf{D}(X_{n+1})$ as the detection subset by applying $\mathbf{D}$ to the uncontaminated feature $X_{n+1}$, which cannot be obtained in practice. Assumption 3.2 says that the detection indicator of each coordinate is independent of each other, which is used to build the connection between $\tilde{\mathcal{O}}_{n+1}$ and $\hat{\mathcal{O}}_{n+1}$, and we will also provide coverage results when this assumption does not hold in Section 5. Next lemma shows that the randomness of $\tilde{\mathcal{O}}_{n+1}$ completely originates from $\hat{\mathcal{O}}_{n+1}$.

**Lemma 3.3.** *Under Assumptions 3.1 and 3.2, it holds that $\tilde{\mathcal{O}}_{n+1} = \hat{\mathcal{O}}_{n+1} \cup \mathcal{O}^*$ and*

$$\check{X}_{n+1}^{\mathrm{DI}} = \mathbf{I}(X_{n+1}, \hat{\mathcal{O}}_{n+1} \cup \mathcal{O}^*). \quad (3)$$

Denote $\hat{\mathcal{O}}_i = \mathbf{D}(X_i)$ for $i = n_0, \ldots, n$ as outputs by applying $\mathbf{D}$ in calibration features. Since $\hat{s}_j(\cdot)$ and $\tau_j$ are independent of calibration set $\mathcal{D}_c$ and $(X_{n+1}, Y_{n+1})$, we know that $\{\hat{\mathcal{O}}_i \cup \mathcal{O}^*\}_{i=n_0}^{n+1}$ are exchangeable. Define the *oracle detection-imputation* (ODI) features

$$\check{X}_i^* = \mathbf{I}(X_i, \hat{\mathcal{O}}_i \cup \mathcal{O}^*), \ i = n_0, \ldots, n+1. \quad (4)$$

By comparing (3) and (4), we have $\check{X}_{n+1}^* = \check{X}_{n+1}^{\mathrm{DI}}$. Let $R_i^* = |Y_i - \hat{\mu}(\check{X}_i^*)|$ be residuals computed by ODI features. We introduce the PI defined as

$$\hat{C}^{\mathrm{ODI}}(\tilde{X}_{n+1}) = \hat{\mu}(\check{X}_{n+1}^{\mathrm{DI}}) \pm \hat{q}_\alpha^+(\{R_i^*\}_{i=n_0}^{n}), \quad (5)$$

and call this oracle method as ODI-CP. For simplicity, we use the absolute residual score to construct PI here, but

our method supports various non-conformity scores as well, such as normalized residual (Lei et al., 2018), conformalized quantile regression (CQR) (Romano et al., 2019), et al. The following proposition provides finite sample coverage for `ODI-CP`.

**Proposition 3.4.** *Suppose the detection procedure* **D** *satisfies Assumptions 3.1 and 3.2, then*

$$\mathbb{P}\left\{Y_{n+1} \in \hat{C}^{\mathrm{ODI}}(\tilde{X}_{n+1})\right\} \geq 1 - \alpha. \quad (6)$$

The role of data-driven mask $\hat{\mathcal{O}}_i$ in `ODI` features (4) is to keep the exchangeability with the processed test feature (3) masked by $\hat{\mathcal{O}}_{n+1}$. This idea is similar to the operations on the calibration set in selective conformal prediction literature (Bao et al., 2024b;a; Jin & Ren, 2025), where they aim to construct PI for selected test data and consider performing a similar selection procedure on the calibration set to satisfy the post-selection exchangeable condition.

### 3.3. Necessity of sure detection property

The next theorem gives a negative result for the case when the sure detection property of **D** does not hold.

**Theorem 3.5.** *If $\check{X}_{n+1}^{\mathrm{DI}}$ contains cellwise outliers, given any PI taking the form $\hat{C}(\check{X}_{n+1}^{\mathrm{DI}}) = \hat{\mu}(\check{X}_{n+1}^{\mathrm{DI}}) \pm \hat{q}_n$ and satisfying marginal coverage $\mathbb{P}\{Y_{n+1} \in \hat{C}(\check{X}_{n+1}^{\mathrm{DI}})\} \geq 1 - \alpha$. For arbitrary $M > 0$, there exists a prediction model $\hat{\mu}$, distributions $P$ and $P_Z$ such that for $(X_{n+1}, Y_{n+1}) \sim P$ and $Z_{n+1} \sim P_Z$, $\mathbb{P}(\hat{q}_n \geq M) \geq 1 - \alpha$. In other words, $\lim_{M \to \infty} \mathbb{E}\left[|\hat{C}(\check{X}_{n+1}^{\mathrm{DI}})|\right] = \infty$.*

This theorem indicates that if outliers persist within the processed feature $\check{X}_{n+1}^{\mathrm{DI}}$, the further PI will fail to provide a meaningful coverage guarantee in a distribution-free and model-free fashion. Therefore, the sure detection property introduced in Assumption 3.1 is reasonable and necessary for predictive inference tasks with the existence of cellwise outliers. The proof of Theorem 3.5 and a similar negative result under CQR score are deferred to Appendix C.2. In fact, Assumption 3.1 can be satisfied in practice if we choose relatively smaller detection thresholds $\{\tau_j\}_{j=1}^d$ in **D** procedure. In fact, a similar sure detection condition also appears in Wasserman & Roeder (2009); Meinshausen et al. (2009); Liu et al. (2022).

## 4. Detect-then-impute conformal prediction

In this section, we develop two practical algorithms named `PDI-CP` and `JDI-CP` to construct the PI, and the corresponding coverage property is analyzed. The `PDI-CP` is based on a proxy of the `ODI` features in the previous section, and the `JDI-CP` is a Jackknife+ type construction to guarantee finite sample coverage.

### 4.1. Proxy detection-imputation

Since $\mathcal{O}^*$ is impossible to know in practice, the best proxy we have access to is the output $\tilde{\mathcal{O}}_{n+1}$ of detection procedure **D**. By replacing $\mathcal{O}^*$ with $\tilde{\mathcal{O}}_{n+1}$ in (4), we have the *proxy detection-imputation* (`PDI`) features

$$\check{X}_i = \mathbf{I}(X_i, \hat{\mathcal{O}}_i \cup \tilde{\mathcal{O}}_{n+1}), \quad i = n_0, \ldots, n. \quad (7)$$

Denote the residuals in calibration set as $\{\check{R}_i := |Y_i - \hat{\mu}(\check{X}_i)|\}_{i=n_0}^n$, and we construct the PI for $Y_{n+1}$ as:

$$\hat{C}^{\mathrm{PDI}}(\tilde{X}_{n+1}) = \hat{\mu}(\check{X}_{n+1}^{\mathrm{DI}}) \pm \hat{q}_\alpha^+(\{\check{R}_i\}_{i=n_0}^n). \quad (8)$$

---

**Algorithm 1** `PDI-CP`

---

**Input:** Calibration set $\{(X_i, Y_i)\}_{i=n_0}^n$, test feature $\tilde{X}_{n+1}$, prediction model $\hat{\mu}$, detection procedure **D**, imputation procedure **I**, miscoverage level $\alpha$.

1: $\tilde{\mathcal{O}}_{n+1} \leftarrow \mathbf{D}(\tilde{X}_{n+1})$
2: $\check{X}_{n+1}^{\mathrm{DI}} \leftarrow \mathbf{I}(\tilde{X}_{n+1}, \tilde{\mathcal{O}}_{n+1})$
3: **for** $i = n_0, \ldots, n$ **do**
4: $\quad \hat{\mathcal{O}}_i \leftarrow \mathbf{D}(X_i)$
5: $\quad \check{X}_i \leftarrow \mathbf{I}(X_i, \hat{\mathcal{O}}_i \cup \tilde{\mathcal{O}}_{n+1})$
6: $\quad \check{R}_i \leftarrow |Y_i - \hat{\mu}(\check{X}_i)|$
7: **end for**
8: $\hat{C}^{\mathrm{PDI}}(\tilde{X}_{n+1}) \leftarrow \hat{\mu}(\check{X}_{n+1}^{\mathrm{DI}}) \pm \hat{q}_\alpha^+(\{\check{R}_i\}_{i=n_0}^n)$

**Output:** $\hat{C}^{\mathrm{PDI}}(\tilde{X}_{n+1})$

---

We call the construction stated in (8) as `PDI-CP`, and summarize its implementation in Algorithm 1. Next, we analyze the coverage property of `PDI-CP`.

**Definition 4.1** ($\ell_1$-sensitivity). The $\ell_1$-sensitivity of prediction model $\hat{\mu}$ is $\sup_{\|x-x'\|_1 \leq 1} |\hat{\mu}(x) - \hat{\mu}(x')| \leq S_{\hat{\mu}}$.

The $\ell_1$-sensitivity measures the stability of a prediction model to its input, which is similar to that in differential privacy literature (Dwork et al., 2014). For example, for a linear regression model $\hat{\mu}(x) = \beta^\top x + b$, whose $\ell_1$-sensitivity is given by $S_{\hat{\mu}} = \|\beta\|_1$.

**Definition 4.2** (Mean Imputation). The imputation function is $\phi_j(\cdot) = \bar{x}_j = \frac{1}{n_0-1}\sum_{i=1}^{n_0-1} X_{ij}$ for $j \in [d]$, i.e., the cellwise sample mean obtained from the training set.

The following theorem provides the coverage property of `PDI-CP` under the Mean Imputation rule.

**Theorem 4.3.** *Suppose the detection procedure* **D** *satisfies Assumptions 3.1 and 3.2, and $\hat{\mu}$ has the $\ell_1$-sensitivity $S_{\hat{\mu}}$ in Definition 4.1. Let $E_i := \max_{j \in [d]} |X_{ij} - \bar{x}_j|$, $i = n_0, \ldots, n, n+1$ be error of Mean Imputation in Definition 4.2. We have*

$$\mathbb{P}\left\{Y_{n+1} \in \hat{C}^{\mathrm{PDI}}(\tilde{X}_{n+1})\right\} \geq 1 - \alpha$$
$$- \left[ F_{R^*}\left(\hat{q}_\alpha^+\left(\{R_i^* + S_{\hat{\mu}} \cdot E_i \cdot |\tilde{\mathcal{O}}_{n+1} \setminus \mathcal{O}^*|\}_{i=n_0}^n\right)\right) \right.$$
$$\left. - F_{R^*}\left(\hat{q}_\alpha^+\left(\{R_i^* - S_{\hat{\mu}} \cdot E_i \cdot |\tilde{\mathcal{O}}_{n+1} \setminus \mathcal{O}^*|\}_{i=n_0}^n\right)\right)\right],$$

where $F_{R^*}$ is the distribution function of `ODI-CP` residuals $\{R_i^*\}_{i=n_0}^{n+1}$.

According to the representation, the coverage gap of `PDI-CP` depends on the error of Mean Imputation and the number of false discoveries $|\tilde{\mathcal{O}}_{n+1} \setminus \mathcal{O}^*|$, which represents the number of inliers incorrectly identified as outliers by the detection procedure $\mathbf{D}$. In an ideal scenario where $\tilde{\mathcal{O}}_{n+1} = \mathcal{O}^*$, `PDI-CP` can achieve the finite sample coverage as `ODI-CP`. The proof for Theorem 4.3 is deferred to Appendix C.3.

### 4.2. Joint detection-imputation

The coverage gaps of `PDI-CP` are caused by the nonexchangeability between `PDI` feature $\check{X}_i$ and processed test feature $\check{X}_{n+1}^{\mathrm{DI}}$. This subsection proposes a robust approach to constructing exchangeable features by applying a modified $\mathbf{DI}$ procedure on test feature $\tilde{X}_{n+1}$, which achieves $1 - 2\alpha$ finite sample coverage in the worst case.

We begin with a careful inspection of *pairwise* exchangeability in test feature and calibration feature after $\mathbf{DI}$. Under Lemma 3.3, we can rewrite `PDI` feature (7) as $\check{X}_i = \mathbf{I}(X_i, \hat{\mathcal{O}}_i \cup \hat{\mathcal{O}}_{n+1} \cup \mathcal{O}^*)$. Since $\hat{\mathcal{O}}_i$ and $\hat{\mathcal{O}}_{n+1}$ are exchangeable, it is easy to verify that $\mathbf{I}(X_i, \hat{\mathcal{O}}_i \cup \hat{\mathcal{O}}_{n+1} \cup \mathcal{O}^*)$ and $\mathbf{I}(X_{n+1}, \hat{\mathcal{O}}_i \cup \hat{\mathcal{O}}_{n+1} \cup \mathcal{O}^*)$ are also exchangeable. Using Lemma 3.3 again, the latter one is identical to $\mathbf{I}(\tilde{X}_{n+1}, \hat{\mathcal{O}}_i \cup \tilde{\mathcal{O}}_{n+1})$. In light of this observation, we have the following pairwise *joint detection-imputation* (`JDI`) features: for $i = n_0, \ldots, n$,

$$\check{X}_{n+1}^i = \mathbf{I}(\tilde{X}_{n+1}, \hat{\mathcal{O}}_i \cup \tilde{\mathcal{O}}_{n+1}),$$
$$\check{X}_i^{n+1} = \mathbf{I}(X_i, \hat{\mathcal{O}}_i \cup \tilde{\mathcal{O}}_{n+1}).$$

Notice that, $\check{X}_i^{n+1}$ is the same as the `PDI` feature (7), and we use a new notation here to emphasize the pairwise relation.

With the pairwise `JDI` features, we leverage the Jackknife+ technique proposed by Barber et al. (2021) to construct the PI for $Y_{n+1}$:

$$\hat{C}^{\mathrm{JDI}}(\tilde{X}_{n+1}) = \big[\hat{q}_\alpha^-(\{\hat{\mu}(\check{X}_{n+1}^i) - \check{R}_i\}_{i=n_0}^n),$$
$$\hat{q}_\alpha^+(\{\hat{\mu}(\check{X}_{n+1}^i) + \check{R}_i\}_{i=n_0}^n)\big], \quad (9)$$

where $\check{R}_i = |Y_i - \hat{\mu}(\check{X}_i^{n+1})| = |Y_i - \hat{\mu}(\check{X}_i)|$ is the same as the residuals in (8). The method presented above is called `JDI-CP`, and we summarize the implementation in Algorithm 2.

**Theorem 4.4.** *Suppose the detection procedure $\mathbf{D}$ satisfies Assumptions 3.1 and 3.2, then*

$$\mathbb{P}\left\{Y_{n+1} \in \hat{C}^{\mathrm{JDI}}(\tilde{X}_{n+1})\right\} \geq 1 - 2\alpha.$$

The proof of Theorem 4.4 is given in Appendix C.4. Even though `JDI-CP` cannot achieve the target level $1 - \alpha$ due

---

**Algorithm 2** `JDI-CP`

**Input:** Same as Algorithm 1.
1: $\tilde{\mathcal{O}}_{n+1} \leftarrow \mathbf{D}(\tilde{X}_{n+1})$
2: **for** $i = n_0, \ldots, n$ **do**
3: $\quad \hat{\mathcal{O}}_i \leftarrow \mathbf{D}(X_i)$
4: $\quad \check{X}_i^{n+1} \leftarrow \mathbf{I}(X_i, \hat{\mathcal{O}}_i \cup \tilde{\mathcal{O}}_{n+1})$
5: $\quad \check{X}_{n+1}^i \leftarrow \mathbf{I}(\tilde{X}_{n+1}, \hat{\mathcal{O}}_i \cup \tilde{\mathcal{O}}_{n+1})$
6: $\quad \check{R}_i \leftarrow |Y_i - \hat{\mu}(\check{X}_i^{n+1})|$
7: **end for**
8: $\hat{Q}_\alpha^- \leftarrow \hat{q}_\alpha^-(\{\hat{\mu}(\check{X}_{n+1}^i) - \check{R}_i\}_{i=n_0}^n)$
9: $\hat{Q}_\alpha^+ \leftarrow \hat{q}_\alpha^+(\{\hat{\mu}(\check{X}_{n+1}^i) + \check{R}_i\}_{i=n_0}^n)$
10: $\hat{C}^{\mathrm{JDI}}(\tilde{X}_{n+1}) \leftarrow [\hat{Q}_\alpha^-, \hat{Q}_\alpha^+]$
**Output:** $\hat{C}^{\mathrm{JDI}}(\tilde{X}_{n+1})$

---

to the intrinsic limit of Jackknife+ type method (Barber et al., 2021), the finite sample result in Theorem 4.4 is still important in practical tasks for two reasons: (1) $\tilde{\mathcal{O}}_{n+1}$ may contain many false discoveries which leads to a large coverage gap for `PDI-CP`; (2) the coverage property of `JDI-CP` holds for arbitrary imputation rules. In Appendix B.2, we provide an alternative version of `JDI-CP`, which achieves a finite sample $1 - \alpha$ coverage guarantee under the same conditions in Theorem 4.4. However, it is quite conservative compared with the original version because it uses $\cup_{i=n_0}^n \hat{\mathcal{O}}_i \cup \tilde{\mathcal{O}}_{n+1}$ to mask calibration features and test feature. In our experiments, `JDI-CP` has a robust coverage control without adjusting the quantile level in (9).

We notice that Jackknife+ technique is also used in CP-MDA-Nested algorithm in Zaffran et al. (2023; 2024) to build PI with missing values in features, and a similar $1 - 2\alpha$ coverage property is proved. However, the algorithmic designs of CP-MDA-Nested and `JDI-CP` are very different. The former uses the observed and exchangeable missing patterns to perform nested masking for each pair of calibration and test features, while `JDI-CP` uses outputs of $\mathbf{D}$ (i.e., $\hat{\mathcal{O}}_i$ and $\tilde{\mathcal{O}}_{n+1}$), which are not exchangeable.

To end this section, we use Figure 1 to illustrate the connection and difference of the proposed methods.

## 5. Coverage analysis beyond isolated detection

The coverage results in the previous section are based on the isolated detection in Assumption 3.2, now we discuss the theoretical properties of our method when the detection score relies on other coordinates, e.g., DDC detection procedure (Rousseeuw & Bossche, 2018). Now we rewrite the detection score $\hat{s}_j(x_j) = \hat{s}_j(x_j; x_{-j})$, where $x_{-j}$ is the subvector of $x$ by dropping $x_j$. Denote

$$\tilde{\mathcal{T}}_{n+1} = \{j \in [d] \setminus \mathcal{O}^* : \hat{s}_j(X_{n+1,j}; \tilde{X}_{n+1,-j}) > \tau_j\},$$
$$\hat{\mathcal{T}}_{n+1} = \{j \in [d] \setminus \mathcal{O}^* : \hat{s}_j(X_{n+1,j}; X_{n+1,-j}) > \tau_j\}.$$

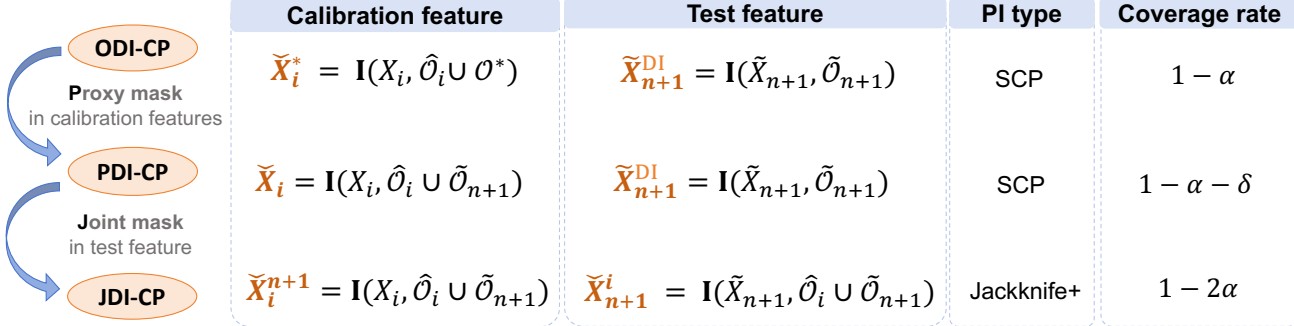

| | Calibration feature | Test feature | PI type | Coverage rate |
|---|---|---|---|---|
| **ODI-CP** | $\check{X}_i^* = \mathbf{I}(X_i, \hat{\mathcal{O}}_i \cup \mathcal{O}^*)$ | $\widetilde{X}_{n+1}^{\mathrm{DI}} = \mathbf{I}(\tilde{X}_{n+1}, \tilde{\mathcal{O}}_{n+1})$ | SCP | $1 - \alpha$ |
| **PDI-CP** | $\check{X}_i = \mathbf{I}(X_i, \hat{\mathcal{O}}_i \cup \tilde{\mathcal{O}}_{n+1})$ | $\widetilde{X}_{n+1}^{\mathrm{DI}} = \mathbf{I}(\tilde{X}_{n+1}, \tilde{\mathcal{O}}_{n+1})$ | SCP | $1 - \alpha - \delta$ |
| **JDI-CP** | $\check{X}_i^{n+1} = \mathbf{I}(X_i, \hat{\mathcal{O}}_i \cup \tilde{\mathcal{O}}_{n+1})$ | $\check{X}_{n+1}^i = \mathbf{I}(\tilde{X}_{n+1}, \hat{\mathcal{O}}_i \cup \tilde{\mathcal{O}}_{n+1})$ | Jackknife+ | $1 - 2\alpha$ |

(Left labels: Proxy mask in calibration features; Joint mask in test feature)

Figure 1. An illustration of the relationship of `ODI-CP`, `PDI-CP` and `JDI-CP`. $\hat{\mathcal{O}}_i = \mathbf{D}(X_i)$ and $\tilde{\mathcal{O}}_{n+1} = \mathbf{D}(\tilde{X}_{n+1})$.

The two sets above represent the false discoveries of $\mathbf{D}$ applied on contaminated feature $\tilde{X}_{n+1}$ and clean feature $X_{n+1}$ except for $j \in \mathcal{O}^*$, respectively. Without Assumption 3.2, the `ODI` features $\{\check{X}_i^*\}_{i=n_0}^n$ and $\check{X}_{n+1}^{\mathrm{DI}}$ are no longer exchangeable, and the coverage guarantee of `ODI-CP` and `JDI-CP` will be broken. The next theorem characterizes the coverage gaps of our algorithms through $\tilde{\mathcal{T}}_{n+1}$ and $\hat{\mathcal{T}}_{n+1}$.

**Theorem 5.1.** *Suppose the detection procedure $\mathbf{D}$ satisfies Assumptions 3.1, and $\hat{\mu}$ has the $\ell_1$-sensitivity $S_{\hat{\mu}}$ in Definition 4.1. Under the Mean Imputation in Definition 4.2,*
*(1)* `PDI-CP` *satisfies that*

$$\mathbb{P}\left\{Y_{n+1} \in \hat{C}^{\mathrm{PDI}}(\tilde{X}_{n+1})\right\} \geq 1 - \alpha$$
$$- \left[F_{R^*}\left(\hat{q}_\alpha^+(\{R_i^*\}_{i=n_0}^n) + \Delta_{n+1}\right) - F_{R^*}(\hat{q}_\alpha^+(\{R_i^*\}_{i=n_0}^n))\right]$$
$$- \left[F_{R^*}\left(\hat{q}_\alpha^+\left(\{R_i^* + S_{\hat{\mu}} \cdot E_i \cdot |\tilde{\mathcal{T}}_{n+1}|\}_{i=n_0}^n\right)\right)\right.$$
$$\left. - F_{R^*}\left(\hat{q}_\alpha^+\left(\{R_i^* - S_{\hat{\mu}} \cdot E_i \cdot |\tilde{\mathcal{T}}_{n+1}|\}_{i=n_0}^n\right)\right)\right], \quad (10)$$

*where $\Delta_{n+1} = S_{\hat{\mu}} \cdot |\tilde{\mathcal{T}}_{n+1} \triangle \hat{\mathcal{T}}_{n+1}| \cdot E_{n+1}$ and $\triangle$ denotes the symmetric difference.*
*(2)* `JDI-CP` *satisfies that*

$$\mathbb{P}\left\{Y_{n+1} \in \hat{C}^{\mathrm{JDI}}(\tilde{X}_{n+1})\right\} \geq 1 - 2\alpha$$
$$- \left[F_Y(\hat{Q}_\alpha^+ + 2\Delta_{n+1}) - F_Y(\hat{Q}_\alpha^+)\right]$$
$$- \left[F_Y(\hat{Q}_\alpha^-) - F_Y(\hat{Q}_\alpha^- - 2\Delta_{n+1})\right],$$

*where $F_Y$ is the distribution function of $Y$, $\hat{Q}_\alpha^-$ and $\hat{Q}_\alpha^+$ are defined in Algorithm 2 where $\check{X}_i^{n+1}$ and $\check{X}_{n+1}^i$ are replaced by $\mathbf{I}(X_i, \hat{\mathcal{O}}_i \cup \mathcal{O}^* \cup \hat{\mathcal{T}}_{n+1})$ and $\mathbf{I}(X_{n+1}, \hat{\mathcal{O}}_i \cup \mathcal{O}^* \cup \hat{\mathcal{T}}_{n+1})$ respectively.*

The proof of Theorem 5.1 is given in Appendix C.5. If the sets $\tilde{\mathcal{T}}_{n+1}$ and $\hat{\mathcal{T}}_{n+1}$ are infinitely close, then the error $\Delta_{n+1} \to 0$ and $\hat{C}^{\mathrm{JDI}}(\tilde{X}_{n+1})$ still has $1 - 2\alpha$ coverage guarantee. Further, if $|\tilde{\mathcal{T}}_{n+1}| = |\hat{\mathcal{T}}_{n+1}| \to 0$, it means the detection method can accurately find the cellwise outliers without false discoveries, then $\hat{C}^{\mathrm{PDI}}(\tilde{X}_{n+1})$ can also achieve the target $1 - \alpha$ coverage.

We use the DDC method as an example to demonstrate the relation between $\tilde{\mathcal{T}}_{n+1}$ and $\hat{\mathcal{T}}_{n+1}$ (see Appendix D for details of DDC method). A commonly used similarity metric of two sets is Jaccard similarity (Murphy, 1996):

$$\mathrm{Jaccard}(\tilde{\mathcal{T}}_{n+1}, \hat{\mathcal{T}}_{n+1}) = \frac{|\tilde{\mathcal{T}}_{n+1} \cap \hat{\mathcal{T}}_{n+1}|}{|\tilde{\mathcal{T}}_{n+1} \cup \hat{\mathcal{T}}_{n+1}|} \in [0, 1]$$

with a value closer to 1 indicating $\tilde{\mathcal{T}}_{n+1}$ and $\hat{\mathcal{T}}_{n+1}$ are more similar. Figure 2 visualizes $\mathrm{Jaccard}(\tilde{\mathcal{T}}_{n+1}, \hat{\mathcal{T}}_{n+1})$ and the number of occurrences of $\tilde{\mathcal{T}}_{n+1} = \hat{\mathcal{T}}_{n+1} = \emptyset$ during 500 trials, each with 100 test points. It can be seen that $\tilde{\mathcal{T}}_{n+1}$ and $\hat{\mathcal{T}}_{n+1}$ are almost the same except in a few trials. In addition, the frequency of $\tilde{\mathcal{T}}_{n+1} = \hat{\mathcal{T}}_{n+1} = \emptyset$ reaches above 0.6.

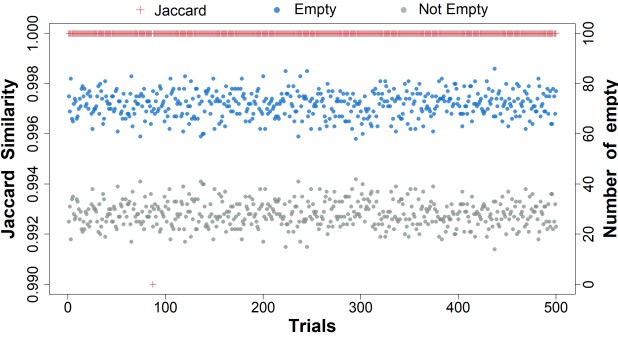

Figure 2. An illustration of the similarity of $\tilde{\mathcal{T}}_{n+1}$ and $\hat{\mathcal{T}}_{n+1}$ (left axis, red cross marks) and the number of occurrences of empty and nonempty sets (right axis, blue and gray dots).

## 6. Simulation

We write $N(\mu, \sigma^2)$ for the normal distribution with mean $\mu$ and variance $\sigma^2$, $SN(\mu, \sigma^2, \alpha)$ for the skewed normal with skewness parameter $\alpha$, $t(k)$ for the $t$-distribution with $k$ degrees of freedom, and $Bern(p)$ for the Bernoulli distribution with success probability $p$. Given any $x \in \mathbb{R}^d$, define $f(x) = \mathbb{E}(Y_i|X_i = x)$ and $\eta_i = Y_i - f(X_i)$. We consider three data generation settings in Lei et al. (2018):

- **Setting A** (linear, homoscedastic, light-tailed): $f(x) = \beta^\top x$; $\{X_{ij}\}_{j=1}^d \overset{i.i.d.}{\sim} N(0,1)$; $\eta_i \sim N(0,1)$; $\eta_i \perp X_i$.

- **Setting B** (nonlinear, homoscedastic, heavy-tailed): $f(x)$ is an additive function of B-splines of $x_1, \ldots, x_d$; $\{X_{ij}\}_{j=1}^d \overset{i.i.d.}{\sim} N(0,1)$; $\eta_i \sim t(2)$; $\eta_i \perp X_i$.

- **Setting C** (linear, correlated features, heteroskedastic, heavy-tailed): $f(x) = \beta^\top x$; $\{X_{ij}\}_{j=1}^d \overset{i.i.d.}{\sim} \frac{1}{3}N(0,1) + \frac{1}{3}SN(0,1,5) + \frac{1}{3}Bern(0.5)$, and then given autocorrelation by redefining sequentially each $X_{ij}$ to be a convex combination of $X_{ij}, X_{i,j-1}, \ldots, X_{i,(j-3)\wedge 1}$; $\eta_i \sim t(2)$, with variable standard deviation $1 + 2\left|f(X_i)\right|^3 / \mathbb{E}\left(|f(X)|^3\right)$.

Let $\epsilon$ be the probability that a cell in the test feature is contaminated. Then we obtain the test feature with entries

$$\tilde{X}_{n+1,j} = \begin{cases} X_{n+1,j}, & \text{with probability } 1-\epsilon \\ Z_{n+1,j}, & \text{with probability } \epsilon \end{cases}, \quad (11)$$

where the noise $Z_{n+1,j} \overset{i.i.d.}{\sim} N(\mu, \sigma)$ and $\mu, \sigma$ are randomly sampled from $U(0,10)$. All simulation results in the following are averages over 200 trials with 200 labeled data and 100 test data. The nominal coverage level is set to be $1-\alpha = 90\%$ and $d=15$. For comparison, we also conduct following methods (see Appendix E.1 for specific forms):

1. `Baseline`: mask the entries of $\{X_i\}_{i=n_0}^n \cup \tilde{X}_{n+1}$ by $\mathcal{O}^*$, then compute residuals on the calibration set and construct SCP interval, which can be seen as an optimal oracle approach to construct split conformal PI in the cellwise outlier case;

2. `SCP`: directly compute residuals on the calibration set and contaminated test feature, then construct SCP interval without performing **DI**;

3. `WCP`: first estimate the likelihood ratio between labeled data and test data using random forests approach, then construct WCP interval.

Figure 3 presents the empirical coverage and length of six methods. As expected, `SCP` and `WCP` fail to reach the target coverage level. Notably, when the contamination probability is high, e.g. $\epsilon = 0.2$, `WCP` even outputs a PI with infinite width. Meanwhile, our proposed method can achieve $1-\alpha$ target coverage and return PIs with almost the same lengths as `Baseline` method. Since `SCP` and `WCP` cannot provide valid coverage control, we regard `Baseline` as the benchmark for the subsequent experiments.

### 6.1. Combinations with other detection methods

This experiment is to verify the validity of our methods under other plausible cellwise detection methods besides

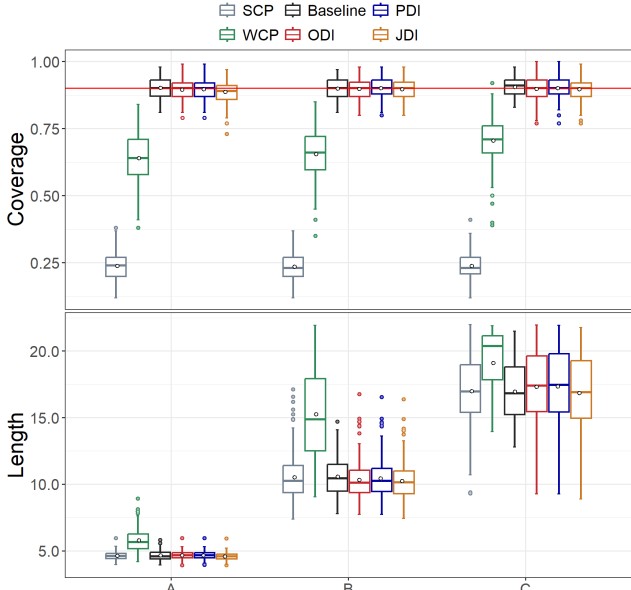

*Figure 3.* Simulation results of six methods. **D** is DDC, **I** is Mean Imputation, $\epsilon = 0.1$ and $\tau_j = \sqrt{\chi^2_{1,0.95}}$. The red line is the target coverage level $1-\alpha = 90\%$.

DDC. Here we consider two procedures: the one-class SVM classifier method (Bates et al., 2023a) with $\tau_j = 0.2$ and the cellMCD estimate method (Raymaekers & Rousseeuw, 2024a) with $\tau_j = \sqrt{\chi^2_{1,0.99}}$, where $\tau_j$ is determined to control the FDR (false discovery rate). The former satisfies Assumption 3.2 while the latter does not. According to Figure 4, our method still has the coverage guarantee when employing other detection procedures. The empirical TPR (true positive rate) and FDR of detection methods are given in Appendix E.2.

### 6.2. Combinations with other imputation methods

Besides Mean Imputation, we also conduct experiments under the other two imputation methods: k-Nearest Neighbour (kNN, Troyanskaya et al. (2001)) and Multivariate Imputation by Chained Equations (MICE, Van Buuren et al. (1999)). Figure 5 shows that if we replace Mean Imputation with kNN or MICE in **DI**, then our method is still able to achieve target $1-\alpha$ coverage.

### 6.3. Performance under different contaminated ratios

From (11), we can see the probability that at least one cell of $X_{n+1}$ is contaminated by $Z_{n+1}$ is $1-(1-\epsilon)^d$, which grows very quickly as $\epsilon$ and $d$ increase. For example, in our setting $d=15$, $\epsilon = 0.05$ suffices to have over $46\%$ of contaminated test data on average, while $\epsilon = 0.2$ can achieve over $97\%$ of contaminated test data. Here we explore the effect of

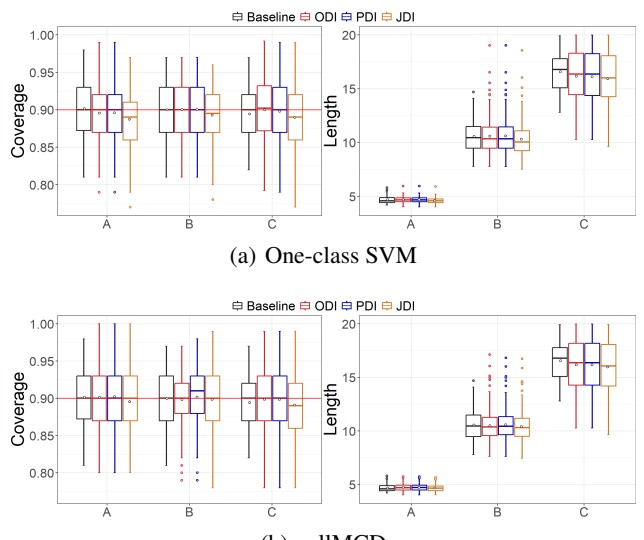

*(a) One-class SVM*

*(b) cellMCD*

*Figure 4.* Simulation results of `Baseline` and our methods with different **D** procedures. **I** is Mean Imputation and $\epsilon = 0.1$.

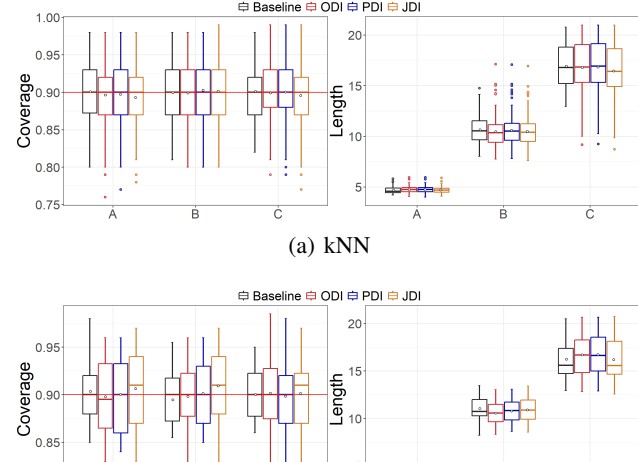

*(a) kNN*

*(b) MICE*

*Figure 5.* Simulation results of `Baseline` and our methods with different **I** procedures. **D** is DDC, $\epsilon = 0.1$ and $\tau_j = \sqrt{\chi^2_{1,0.95}}$.

contamination levels $\epsilon$ on our method. We set **D** as DDC, and the parameter $p$ in detection threshold $\tau_j = \sqrt{\chi^2_{1,p}}$ as $\{0.95, 0.93, 0.9\}$ corresponding to $\epsilon \in \{0.1, 0.15, 0.2\}$ to control FDR. According to Figure 6, our method can achieve a very tight coverage control under each contamination level. Other supplementary simulation results refer to Appendix E.3.

### 6.4. Performance under different detection thresholds

We conduct this experiment by varying the DDC detection threshold $\sqrt{\chi^2_{1,p}}$ (adjusting $p$). As the threshold decreases, the number of discoveries increases, leading to a higher FDR and a higher TPR. Table 1 shows that our method has a robust coverage performance when the threshold changes and can achieve approximate control of coverage when TPR<1.

*Table 1.* Empirical TPR and FDR of DDC with different thresholds under Setting A when $\epsilon = 0.1$.

| $p$ | 0.99 | 0.9 | 0.7 | 0.5 |
|---|---|---|---|---|
| TPR | 0.987 | 0.992 | 0.995 | 1 |
| FDR | 0.035 | 0.340 | 0.669 | 0.793 |
| PDI coverage | 0.902 | 0.901 | 0.907 | 0.909 |
| JDI coverage | 0.895 | 0.899 | 0.904 | 0.904 |

### 6.5. Performance with CQR score

To demonstrate that our method can be combined with other non-conformity scores, we conduct an experiment using

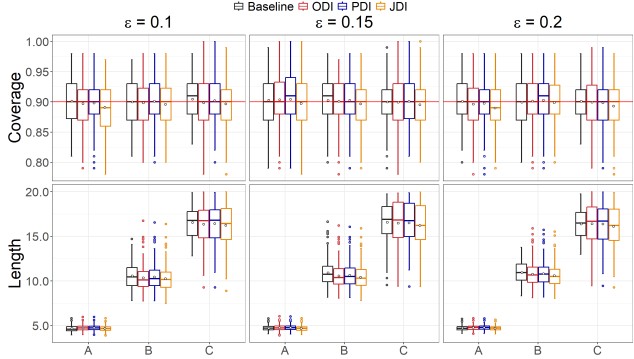

*Figure 6.* Simulation results of `Baseline` and our methods when $\epsilon \in \{0.1, 0.15, 0.2\}$. **D** is DDC and **I** is Mean Imputation.

CQR to construct PIs. We compare the `Baseline` with our method under Setting A with different contaminated ratios. Figure 7 shows that the PIs constructed by our method with CQR score can still approximately achieve the target coverage rate.

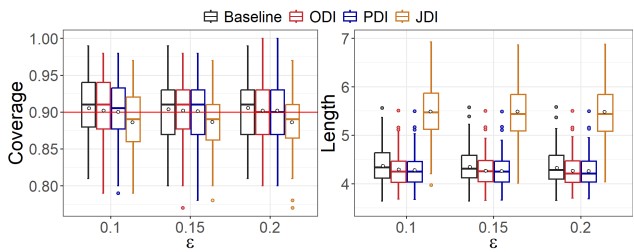

*Figure 7.* Simulation results of `Baseline` and our methods when PI is constructed by CQR. **D** is DDC and **I** is Mean Imputation.

# 7. Application on real data

## 7.1. Airfoil data

We apply the proposed method to the airfoil dataset from the UCI Machine Learning Repository (Dua & Graff, 2019), where the response $Y$ and covariates $X$ (with 5 dimensions) are described in Appendix E.4. We select 1000 labeled data and 500 test data in 100 trials. Since it is unknown which cells are outliers in reality, we artificially introduce outliers with $\epsilon = 0.02$ to construct test features with both genuine and artificial cellwise outliers. The details of the experiment are presented in Appendix E.4. Figure 8 shows that our method can achieve $1 - \alpha$ coverage while SCP and WCP fail to reach the target coverage.

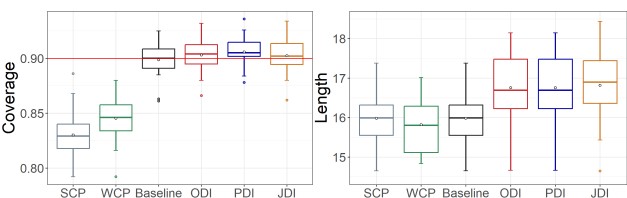

*Figure 8.* Experimental results on airfoil dataset. **D** is DDC and **I** is Mean Imputation.

## 7.2. Wind direction data

Another example involves the hourly wind direction data from a meteorological station in the Central-West region of Brazil (https://tempo.inmet.gov.br/TabelaEstacoes/A001). In 100 trials, we randomly select 1000 labeled data and 500 test data, and add artificial outliers with $\epsilon = 0.02$ to test features. The variable list and data processing steps are presented in Appendix E.5. From Figure 9, we conclude that our method outperforms SCP and WCP in actual cases where cellwise outliers are present in the test feature.

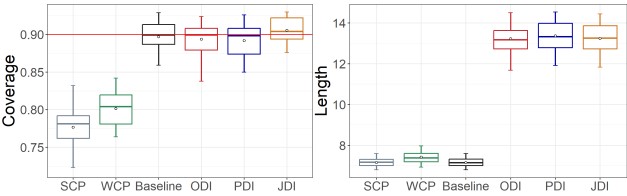

*Figure 9.* Experimental results on wind direction dataset. **D** is DDC and **I** is Mean Imputation.

## 7.3. Riboflavin data

To further demonstrate robustness, we test our method on the gene expression dataset for riboflavin production provided by DSM (Kaiseraugst, Switzerland), which was offered by Bühlmann & Mandozzi (2014) and confirmed to have cellwise outliers by Liu et al. (2022). The details can be

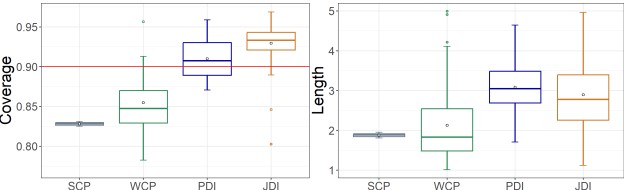

*Figure 10.* Experimental results on riboflavin dataset. **D** is DDC and **I** is Mean Imputation.

found in Appendix E.6. Figure 10 shows that our methods maintain coverage above the target $1 - \alpha = 90\%$ while WCP fails to provide meaningful PIs, that is, the PIs constructed by WCP may have infinite length.

# 8. Conclusion

This paper proposes a new detect-then-impute conformal prediction framework to address the cellwise outliers in the test feature. We develop two efficient algorithms PDI-CP and JDI-CP to construct prediction intervals, which can be wrapped around arbitrary mainstream detection and imputation procedures. In particular, JDI-CP achieves a finite sample $1 - 2\alpha$ coverage guarantee under the sure detection property of the detection procedure. We conduct extensive experiments to illustrate the robustness and efficiency of our algorithms in both synthetic data and real applications.

# Acknowledgements

We would like to thank the anonymous reviewers and area chair for their helpful comments. Changliang Zou was supported by the National Key R&D Program of China (Grant Nos. 2022YFA1003800, 2022YFA1003703), and the National Natural Science Foundation of China (Grant No. 12231011). Haojie Ren was supported by the National Key R&D Program of China (Grant No. 2024YFA1012200), and the National Natural Science Foundation of China (Grant No. 12471262), Young Elite Scientists Sponsorship Program by CAST and Shanghai Jiao Tong University 2030 Initiative. Yajie Bao was supported by the Postdoctoral Fellowship Program of CPSF (Grant No. GZC20251996), and the fellowship from CPSF (Grant No. 2025M773046).

# Impact Statement

This paper presents the work to advance the field of conformal prediction, which promotes the reliability of machine learning. The tools introduced in this study can be utilized across a wide range of machine learning applications, including those with societal implications, but they do not directly implicate any specific ethical or impact-related concerns, so there is nothing we feel must be specifically highlighted.

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

# A. Notation table

Table 2. The mathematical notations in this paper.

| Notation | Meaning | Comments |
|---|---|---|
| $X_i$ | The $i$th instance | Data model |
| $Y_i$ | The label corresponding to $X_i$ | Data model |
| $\tilde{X}_{n+1}$ | The test feature with cellwise outliers | Data model |
| $Z_{n+1}$ | The instance of outliers with arbitrary distribution | Data model |
| $\check{X}_{n+1}^{\text{DI}}$ | The test feature processed by **DI** | Data model |
| $X_{ij}$ | The $j$th coordinate of the $i$th instance | Data model |
| $x$ | The vector $x = (x_1, x_2, ..., x_d)^T \in \mathbb{R}^d$ | Data model |
| $x_\ell$ | The $\ell$th coordinate of $x$ | Data model |
| $\check{X}_i^*$ | The `ODI` feature | Data model |
| $\check{X}_i$ | The `PDI` feature | Data model |
| $\check{X}_{n+1}^i$ | The `JDI` feature | Data model |
| $\check{X}_i^{n+1}$ | The `JDI` feature which is the same as the `PDI` feature $\check{X}_i$ | Data model |
| $R_i^*$ | The residual computed by `ODI` features | Algorithm output |
| $\check{R}_i$ | The residual computed by `PDI` features | Algorithm output |
| $\mathcal{D}_l$ | The labeled dataset | Set |
| $\mathcal{D}_t$ | The training dataset | Set |
| $\mathcal{D}_c$ | The calibration dataset | Set |
| $\mathcal{O}^*$ | The coordinates of cellwise outliers in $\tilde{X}_{n+1}$ | Set |
| $\hat{\mathcal{O}}_i$ | The detected coordinates by applying **D** to $X_i$ | Set |
| $\tilde{\mathcal{O}}_{n+1}$ | The detected coordinates by applying **D** to $\tilde{X}_{n+1}$ | Set |
| $\tilde{\mathcal{T}}_{n+1}$ | The false discoveries of **D** applied on $\tilde{X}_{n+1}$ except for $j \in \mathcal{O}^*$ | Set |
| $\hat{\mathcal{T}}_{n+1}$ | The false discoveries of **D** applied on $X_{n+1}$ except for $j \in \mathcal{O}^*$ | Set |
| $\hat{\mu}(\cdot)$ | The prediction model | Function |
| $\hat{q}_\alpha^-(\cdot)$ | The $\lceil \alpha(n+1) \rceil$ quantile function | Function |
| $\hat{q}_\alpha^+(\cdot)$ | The $\lceil (1-\alpha)(n+1) \rceil$ quantile function | Function |
| $\hat{s}_j(\cdot)$ | The cellwise score function of $j$th coordinate | Function |
| $\phi_j(\cdot)$ | The imputation function of $j$th coordinate | Function |
| $\mathbf{D}(x)$ | The output of detection procedure | Function |
| $\mathbf{I}(x, \mathcal{O})$ | The output of imputation procedure | Function |
| $F_{R^*}$ | The distribution function of `ODI-CP` residuals $\{R_i^*\}_{i=n_0}^{n+1}$ | Function |
| $n_0$ | The critical point of split | Algorithm parameter |
| $\alpha$ | The target coverage level | Algorithm parameter |
| $\tau_j$ | The detection threshold of $j$th coordinate | Algorithm parameter |
| $N(\mu, \sigma^2)$ | The normal distribution with mean $\mu$ and variance $\sigma^2$ | Distribution model |
| $SN(\mu, \sigma^2, \alpha)$ | The skewed normal with skewness parameter $\alpha$ | Distribution model |
| $t(k)$ | The $t$-distribution with $k$ degrees of freedom | Distribution model |
| $Bern(p)$ | The Bernoulli distribution with success probability $p$ | Distribution model |

# B. More methods to construct PI

## B.1. Naive combination of DI and CP

We consider a method by naively combining the DI procedure with split conformal prediction, which is abbreviated as `Naive-DI`. The processed features of `Naive-DI` are defined as

$$\check{X}_i^{\texttt{Naive-DI}} = \mathbf{I}(X_i, \tilde{\mathcal{O}}_{n+1}), \quad i = n_0, \dots, n+1,$$

and the PI for $Y_{n+1}$:

$$\hat{C}^{\texttt{Naive-DI}}(\tilde{X}_{n+1}) = \hat{\mu}(\check{X}_{n+1}^{\texttt{Naive-DI}}) \pm \hat{q}_\alpha^+(\{|Y_i - \hat{\mu}(\check{X}_i^{\texttt{Naive-DI}})|\}_{i=n_0}^n). \tag{B.1}$$

However, `Naive-DI` constructed in this way fails to achieve $1 - \alpha$ coverage in several cases. As an example, we consider the coverage of PIs constructed by `ODI-CP`, `PDI-CP` and `Naive-DI` under Setting A when $\epsilon = \{0.1, 0.15, 0.2\}$, $\tau_j = \sqrt{\chi_{1,0.9}^2}$ and $Z_{n+1,j} \overset{i.i.d.}{\sim} N(\mu, \sigma)$ where $\mu, \sigma$ are randomly sampled from $U(0, 0.1)$. From Figure 11, it can be seen that when the gap between $Z_{n+1}$ and the clean data is not very large, and the detection threshold $\tau_j$ is small enough to detect more outliers, the `Naive-DI` method fails to $1 - \alpha$ coverage regardless of the level of contaminated probability. In comparison, our method has a better coverage in these cases, which is due to the consideration of the effect of $\{\hat{\mathcal{O}}_i\}_{i=n_0}^n$ when masking. This empirical evidence confirms that simply combining DI with CP (without our proposed modifications) cannot guarantee valid coverage.

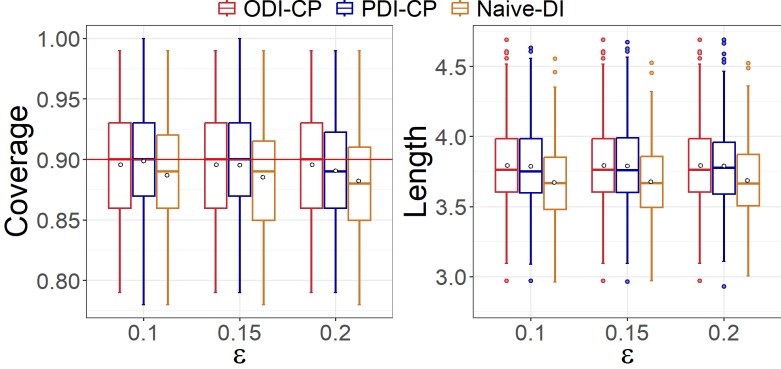

*Figure 11.* Simulation results of `ODI-CP`, `PDI-CP` and `Naive-DI` under Setting A when $\epsilon \in \{0.1, 0.15, 0.2\}$. **D** is DDC, **I** is Mean Imputation.

### B.2. An alternative version of JDI

If we take into account the cellwise outliers contained in all calibration features, we can provide the following conservative version of `JDI`, whose features and PI are listed below.

Conservative-JDI (`C-JDI`) features:

$$\check{X}_i^{\text{C-JDI}} = \mathbf{I}(X_i, \bigcup_{i=n_0}^n \hat{\mathcal{O}}_i \cup \tilde{\mathcal{O}}_{n+1}), \quad i = n_0, \ldots, n+1,$$

and the PI for $Y_{n+1}$:

$$\hat{C}^{\text{C-JDI}}(\tilde{X}_{n+1}) = \hat{\mu}(\check{X}_{n+1}^{\text{C-JDI}}) \pm \hat{q}_\alpha^+(\{|Y_i - \hat{\mu}(\check{X}_i^{\text{C-JDI}})|\}_{i=n_0}^n). \tag{B.2}$$

We introduce Theorem B.1 to demonstrate the coverage guarantee of `C-JDI`.

**Theorem B.1.** *Suppose the detection procedure* **D** *satisfies Assumptions 3.1 and 3.2, then*

$$\mathbb{P}\left\{Y_{n+1} \in \hat{C}^{\text{C-JDI}}(\tilde{X}_{n+1})\right\} \geq 1 - \alpha.$$

*Proof.* Notice that, Lemma 3.3 shows that $\tilde{\mathcal{O}}_{n+1} = \hat{\mathcal{O}}_{n+1} \cup \mathcal{O}^*$, which implies that $\bigcup_{i=n_0}^{n+1} \hat{\mathcal{O}}_i \cup \mathcal{O}^* = \bigcup_{i=n_0}^n \hat{\mathcal{O}}_i \cup \tilde{\mathcal{O}}_{n+1}$. Given the unordered set of $\{X_i\}_{i=n_0}^{n+1}$, the joint mask $\bigcup_{i=n_0}^{n+1} \hat{\mathcal{O}}_i \cup \mathcal{O}^*$ is fixed. Hence we know $\{\check{X}_i^{\text{C-JDI}}\}_{i=n_0}^{n+1}$ are exchangeable. Using the standard proof of split conformal prediction (Lei et al., 2018; Tibshirani et al., 2019), we can prove the result. $\square$

We present an example for comparison of `PDI-CP`, `JDI-CP`, and `C-JDI` in three settings, where $\epsilon = 0.05$, $\tau_j$ and $Z_{n+1}$ are the same as those in Appendix B.1. From Figure 12, we can see that our method can achieve the target coverage rate, which is consistent with simulation results. Although the `C-JDI` method can also achieve the target coverage, its PI is wider and coverage rate is looser than `PDI-CP` and `JDI-CP`, indicating that the PIs constructed by `C-JDI` are more conservative. Taking into account both Appendix B.1 and B.2, our method outperforms `Naive-DI` and `C-JDI`, which is able to build tight PIs that satisfy the target coverage.

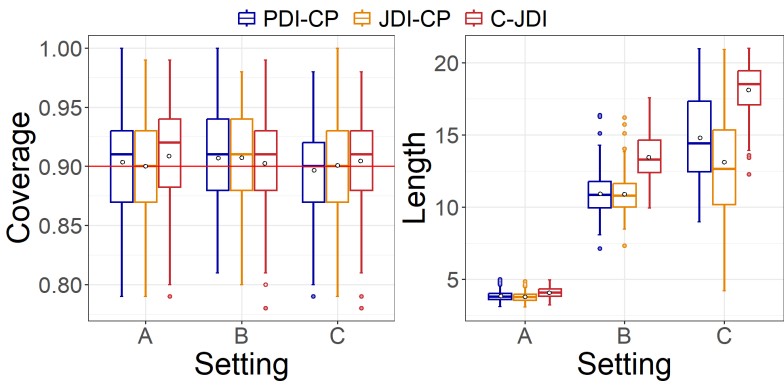

*Figure 12.* Simulation results of `PDI`, `JDI` and `C-JDI`. **D** is DDC, **I** is Mean Imputation and $\epsilon = 0.05$.

## C. Technical Proofs

In this section, we fix the training set $\{(X_i, Y_i)\}_{i=1}^{n_0-1}$ and only consider the randomness from calibration set $\{(X_i, Y_i)\}_{i=n_0}^{n}$ and test data $(X_{n+1}, Y_{n+1}, Z_{n+1})$.

### C.1. Proof of Lemma 3.3 and Proposition 3.4

*Proof.* Firstly, we prove Lemma 3.3. Notice that for any $j \in [d] \setminus \mathcal{O}^*$, we have $\tilde{X}_{n+1,j} = X_{n+1,j}$. It follows that

$$
\begin{aligned}
\tilde{\mathcal{O}}_{n+1} &= \mathcal{O}^* \cup \{j \in [d] \setminus \mathcal{O}^* : \hat{s}_j(\tilde{X}_{n+1,j}) > \tau_j\} \\
&= \mathcal{O}^* \cup \{j \in [d] \setminus \mathcal{O}^* : \hat{s}_j(X_{n+1,j}) > \tau_j\} \\
&= \mathcal{O}^* \cup \{j \in \mathcal{O}^* : \hat{s}_j(X_{n+1,j}) > \tau_j\} \cup \{j \in [d] \setminus \mathcal{O}^* : \hat{s}_j(X_{n+1,j}) > \tau_j\} \\
&= \mathcal{O}^* \cup \hat{\mathcal{O}}_{n+1},
\end{aligned}
$$

where the first equality holds due to Assumption 3.1; and the second equality holds due to Assumption 3.2. From the definition, we know

$$
[\mathbf{I}(\tilde{X}_{n+1}, \hat{\mathcal{O}}_{n+1} \cup \mathcal{O}^*)]_j = \begin{cases} \tilde{X}_{n+1,j} = X_{n+1,j}, & j \notin \hat{\mathcal{O}}_{n+1} \cup \mathcal{O}^* \\ \phi_j(\{\tilde{X}_{n+1,l}\}_{l \notin \hat{\mathcal{O}}_{n+1} \cup \mathcal{O}^*}) = \phi_j(\{X_{n+1,l}\}_{l \notin \hat{\mathcal{O}}_{n+1} \cup \mathcal{O}^*}), & j \in \hat{\mathcal{O}}_{n+1} \cup \mathcal{O}^* \end{cases}
$$
$$
= [\mathbf{I}(X_{n+1}, \hat{\mathcal{O}}_{n+1} \cup \mathcal{O}^*)]_j.
$$

Thus we have

$$
\check{X}_{n+1}^{\mathrm{DI}} = \mathbf{I}(\tilde{X}_{n+1}, \tilde{\mathcal{O}}_{n+1}) = \mathbf{I}(\tilde{X}_{n+1}, \hat{\mathcal{O}}_{n+1} \cup \mathcal{O}^*) = \mathbf{I}(X_{n+1}, \hat{\mathcal{O}}_{n+1} \cup \mathcal{O}^*), \tag{C.1}
$$

which is exchangeable to $\check{X}_i^* = \mathbf{I}(X_i, \hat{\mathcal{O}}_i \cup \mathcal{O}^*)$ for $i = n_0, \ldots, n$.

Next, we prove Proposition 3.4. Recalling that $R_i^* = |Y_i - \hat{\mu}(\check{X}_i^*)|$ for $i = n_0, \ldots, n$. Denote $R_{n+1}^* = |Y_{n+1} - \hat{\mu}(\check{X}_{n+1}^{\mathrm{DI}})|$. By the exchangeability of $\{R_i^*\}_{i=n_0}^{n+1}$, we can guarantee

$$
\mathbb{P}\left\{Y_{n+1} \in \hat{C}^{\mathrm{ODI}}(\tilde{X}_{n+1})\right\} = \mathbb{P}\left\{R_{n+1}^* \leq \hat{q}_\alpha^+(\{R_i^*\}_{i=n_0}^n)\right\} \geq 1 - \alpha. \tag{C.2}
$$

$\square$

## C.2. Proof of Theorem 3.5

*Proof.* We consider the case where $d = 2$. Suppose $X$ is a two-dimensional standard normal random variable with $n$ observations, the corresponding labels are $Y_i = X_{i,1} + X_{i,2}$ where $X_{i,1}, X_{i,2} \sim \text{Uniform}([0,1])$ for $i \in [n]$. Suppose $\hat{\mu}$ is a linear regression model $\hat{\mu}(x) = \beta_1 x_1 + \beta_2 x_2$ where $\beta_2 \neq 0$. The test point $\tilde{X}_{n+1} = (X_{n+1,1}, X_{n+1,2})^\top$ where $X_{n+1,2} = \frac{M+1}{\beta_2} \mathbb{1}\{\beta_1 \geq 1\} + \frac{M-\beta_1+2}{\beta_2} \mathbb{1}\{\beta_1 < 1\}$ for some large positive value $M$.

Now we consider the case $\check{X}_{n+1}^{\text{DI}}$ still contains cellwise outlier after **DI**, that is $\mathbf{D}(\tilde{X}_{n+1}) = \emptyset$ and $\check{X}_{n+1}^{\text{DI}} = \tilde{X}_{n+1}$. When $\beta_1 \geq 1$, notice that

$$|Y_{n+1} - \hat{\mu}(\check{X}_{n+1}^{\text{DI}})| = |(1 - \beta_1)X_{n+1,1} + X_{n+1,2} - M - 1| \geq M. \tag{C.3}$$

By the form of PI, it holds that

$$1 - \alpha = \mathbb{P}\{Y_{n+1} \in \hat{C}(\check{X}_{n+1}^{\text{DI}})\} = \mathbb{P}\left\{|Y_{n+1} - \hat{\mu}(\check{X}_{n+1}^{\text{DI}})| \leq \hat{q}_n\right\} \leq \mathbb{P}(\hat{q}_n \geq M).$$

When $\beta_1 < 1$, notice that

$$|Y_{n+1} - \hat{\mu}(\check{X}_{n+1}^{\text{DI}})| = |(1 - \beta_1)X_{n+1,1} + X_{n+1,2} - M - 2 + \beta_1| \geq M,$$

and (C.3) still holds.

$\square$

**Theorem C.1.** *If $\check{X}_{n+1}^{\text{DI}}$ contains cellwise outliers, given any PI taking the form $\hat{C}(\check{X}_{n+1}^{\text{DI}}) = [\hat{f}^{lo}(\check{X}_{n+1}^{\text{DI}}) - \hat{q}_n, \hat{f}^{up}(\check{X}_{n+1}^{\text{DI}}) + \hat{q}_n]$, where $\hat{f}^{lo}$ and $\hat{f}^{up}$ are the lower and upper quantile regression models, and $\hat{q}_n$ is the quantile of the empirical distribution of CQR score computed on the calibration set. Suppose $\hat{C}(\check{X}_{n+1}^{\text{DI}})$ satisfies marginal coverage $\mathbb{P}\{Y_{n+1} \in \hat{C}(\check{X}_{n+1}^{\text{DI}})\} \geq 1 - \alpha$. For arbitrary $M > 0$, there exists $\hat{f}^{lo}$ and $\hat{f}^{up}$, and distributions $P$ and $P_Z$ such that for $(X_{n+1}, Y_{n+1}) \sim P$ and $Z_{n+1} \sim P_Z$, $\mathbb{P}(\hat{q}_n \geq M) \geq 1 - \alpha$. In other words, $\lim_{M \to \infty} \mathbb{E}\left[|\hat{C}(\check{X}_{n+1}^{\text{DI}})|\right] = \infty$.*

*Proof.* Following the proof of Theorem 3.5 in Appendix B.2, the labels are $Y_i = X_{i,1} + X_{i,2}$ where $X_{i,1}, X_{i,2} \sim \text{Uniform}([0,1])$ for $i \in [n+1]$. Suppose $\hat{f}^{lo}(x) = \beta_1^{lo} x_1 + \beta_2^{lo} x_2$ where $\beta_2^{lo} \neq 0$, and the test point $\tilde{X}_{n+1} = (X_{n+1,1}, Z_{n+1,2})^\top$ where the outlier is given by

$$Z_{n+1,2} = \frac{M+1}{\beta_2^{lo}} \mathbb{1}\{\beta_1^{lo} \geq 1\} + \frac{M+2}{\beta_2^{lo}} \mathbb{1}\{0 < \beta_1^{lo} < 1\} + \frac{M - \beta_1^{lo} + 2}{\beta_2^{lo}} \mathbb{1}\{\beta_1^{lo} \leq 0\}$$

for some large positive value $M$. If $\check{X}_{n+1}^{\text{DI}}$ still contains $Z_{n+1,2}$ and $\hat{C}(\check{X}_{n+1}^{\text{DI}})$ covers the true label, we have

$$\max\{\hat{f}^{lo}(\check{X}_{n+1}^{\text{DI}}) - Y_{n+1}, Y_{n+1} - \hat{f}^{up}(\check{X}_{n+1}^{\text{DI}})\} \geq \hat{f}^{lo}(\check{X}_{n+1}^{\text{DI}}) - Y_{n+1} \geq M,$$

which means $\mathbb{P}(\hat{q}_n \geq M) \geq \mathbb{P}(Y_{n+1} \in \hat{C}(\check{X}_{n+1}^{\text{DI}})) \geq 1 - \alpha.$ $\square$

## C.3. Proof of Theorem 4.3

*Proof.* According to Definition 4.2, we denote by $\bar{x}_i = (\bar{x}_1, \ldots, \bar{x}_j, \ldots, \bar{x}_d)^\top$ the Mean Imputation values of $X_i$ for $i = n_0, \ldots, n+1$. Denote

$$\hat{\delta}_{i,j} = \mathbb{1}\{\hat{s}(X_{i,j}) \leq \tau_j\}, \quad i = n_0, \ldots, n+1, \ j \in [d], \tag{C.4}$$

and

$$\hat{\Delta}_i = \text{diag}(\{\hat{\delta}_{i,j}\}_{j \in [d]}), \quad i = n_0, \ldots, n+1, \tag{C.5}$$

$$\tilde{\Delta}_{n+1} = \text{diag}(\{\tilde{\delta}_{n+1,j}\}_{j \in [d]}), \tag{C.6}$$

$$\Delta_{n+1} = \text{diag}(\{\delta_{n+1,j}\}_{j \in [d]}),$$

where $\tilde{\delta}_{n+1,j} = \mathbb{1}\{\hat{s}_j(\tilde{X}_{n+1,j}) \le \tau_j\}$ and $\delta_{n+1,j} = \mathbb{1}\{j \in [d] \setminus \mathcal{O}^*\}$. Then the difference between ODI feature and PDI feature can be written as

$$
\begin{aligned}
\check{X}_i - \check{X}_i^* &= \left[ \hat{\Delta}_i \tilde{\Delta}_{n+1} X_i + (I - \hat{\Delta}_i \tilde{\Delta}_{n+1}) \bar{\boldsymbol{x}}_i \right] - \left[ \hat{\Delta}_i \Delta_{n+1} X_i + (I - \hat{\Delta}_i \Delta_{n+1}) \bar{\boldsymbol{x}}_i \right] \\
&= \hat{\Delta}_i (\tilde{\Delta}_{n+1} - \Delta_{n+1}) X_i - \hat{\Delta}_i (\tilde{\Delta}_{n+1} - \Delta_{n+1}) \bar{\boldsymbol{x}}_i \\
&= \hat{\Delta}_i (\tilde{\Delta}_{n+1} - \Delta_{n+1})(X_i - \bar{\boldsymbol{x}}_i).
\end{aligned}
\tag{C.7}
$$

Denote by $E_{i,j} = |X_{i,j} - \bar{x}_j|$ the Mean Imputation error of each entry for $i = n_0, \dots, n$. Invoking $\ell_1$-sensitivity of $\hat{\mu}$, we have

$$
\begin{aligned}
\left| \check{R}_i - R_i^* \right| &= \left| |Y_i - \hat{\mu}(\check{X}_i)| - |Y_i - \hat{\mu}(\check{X}_i^*)| \right| \\
&\le \left| \hat{\mu}(\check{X}_i) - \hat{\mu}(\check{X}_i^*) \right| \le S_{\hat{\mu}} \cdot \left\| \check{X}_i - \check{X}_i^* \right\|_1 \\
&= S_{\hat{\mu}} \cdot \sum_{j=1}^d \hat{\delta}_{i,j} |\tilde{\delta}_{n+1,j} - \delta_{n+1,j}| |X_{i,j} - \bar{x}_j| \\
&= S_{\hat{\mu}} \cdot \sum_{j \in [d] \setminus \mathcal{O}^*} \hat{\delta}_{i,j} (\delta_{n+1,j} - \tilde{\delta}_{n+1,j}) \cdot E_{i,j} \\
&\le S_{\hat{\mu}} \cdot \max_{j \in [d]} E_{i,j} \cdot \sum_{j \in [d] \setminus \mathcal{O}^*} (\delta_{n+1,j} - \tilde{\delta}_{n+1,j}) \\
&:= S_{\hat{\mu}} \cdot E_i \cdot |\tilde{\mathcal{O}}_{n+1} \setminus \mathcal{O}^*|,
\end{aligned}
\tag{C.8}
$$

where $E_i := \max_{j \in [d]} E_{i,j}$ and the third equality holds due to Assumption 3.1.

By the construction of $\hat{C}^{\mathrm{ODI}}(\tilde{X}_{n+1})$ in (5), we have

$$
\begin{aligned}
\mathbb{P}\left\{ Y_{n+1} \in \hat{C}^{\mathrm{PDI}}(\tilde{X}_{n+1}) \right\} &= \mathbb{P}\left\{ |Y_{n+1} - \hat{\mu}(\check{X}_{n+1}^{\mathrm{DI}})| \le \hat{q}_\alpha^+(\{\check{R}_i\}_{i=n_0}^n) \right\} \\
&= \mathbb{P}\left\{ R_{n+1}^* \le \hat{q}_\alpha^+(\{\check{R}_i\}_{i=n_0}^n) \right\},
\end{aligned}
\tag{C.9}
$$

and the coverage gap between $\hat{C}^{\mathrm{ODI}}(\tilde{X}_{n+1})$ and $\hat{C}^{\mathrm{PDI}}(\tilde{X}_{n+1})$ is

$$
\begin{aligned}
&\left| \mathbb{P}\left\{ Y_{n+1} \in \hat{C}^{\mathrm{PDI}}(\tilde{X}_{n+1}) \right\} - \mathbb{P}\left\{ Y_{n+1} \in \hat{C}^{\mathrm{ODI}}(\tilde{X}_{n+1}) \right\} \right| \\
&= \left| \mathbb{P}\left\{ R_{n+1}^* \le \hat{q}_\alpha^+(\{\check{R}_i\}_{i=n_0}^n) \right\} - \mathbb{P}\left\{ R_{n+1}^* \le \hat{q}_\alpha^+(\{R_i^*\}_{i=n_0}^n) \right\} \right| \\
&= \mathbb{P}\left\{ \min\left( \hat{q}_\alpha^+(\{\check{R}_i\}_{i=n_0}^n), \hat{q}_\alpha^+(\{R_i^*\}_{i=n_0}^n) \right) < R_{n+1}^* \le \max\left( \hat{q}_\alpha^+(\{\check{R}_i\}_{i=n_0}^n), \hat{q}_\alpha^+(\{R_i^*\}_{i=n_0}^n) \right) \right\} \\
&\le \mathbb{P}\left\{ \hat{q}_\alpha^+\left( \{R_i^* - S_{\hat{\mu}} \cdot E_i \cdot |\tilde{\mathcal{O}}_{n+1} \setminus \mathcal{O}^*|\}_{i=n_0}^n \right) < R_{n+1}^* \le \hat{q}_\alpha^+\left( \{R_i^* + S_{\hat{\mu}} \cdot E_i \cdot |\tilde{\mathcal{O}}_{n+1} \setminus \mathcal{O}^*|\}_{i=n_0}^n \right) \right\} \\
&= F_{R^*}\left( \hat{q}_\alpha^+\left( \{R_i^* + S_{\hat{\mu}} \cdot E_i \cdot |\tilde{\mathcal{O}}_{n+1} \setminus \mathcal{O}^*|\}_{i=n_0}^n \right) \right) - F_{R^*}\left( \hat{q}_\alpha^+\left( \{R_i^* - S_{\hat{\mu}} \cdot E_i \cdot |\tilde{\mathcal{O}}_{n+1} \setminus \mathcal{O}^*|\}_{i=n_0}^n \right) \right),
\end{aligned}
\tag{C.10}
$$

where $F_{R^*}$ is the distribution function of ODI-CP residuals $\{R_i^*\}_{i=n_0}^{n+1}$. Combined with Assumption 3.2 and Proposition 3.4, we can obtain the coverage property of PDI-CP

$$
\begin{aligned}
\mathbb{P}\left\{ Y_{n+1} \in \hat{C}^{\mathrm{PDI}}(\tilde{X}_{n+1}) \right\} &\ge 1 - \alpha \\
&- \left[ F_{R^*}\left( \hat{q}_\alpha^+\left( \{R_i^* + S_{\hat{\mu}} \cdot E_i \cdot |\tilde{\mathcal{O}}_{n+1} \setminus \mathcal{O}^*|\}_{i=n_0}^n \right) \right) - F_{R^*}\left( \hat{q}_\alpha^+\left( \{R_i^* - S_{\hat{\mu}} \cdot E_i \cdot |\tilde{\mathcal{O}}_{n+1} \setminus \mathcal{O}^*|\}_{i=n_0}^n \right) \right) \right].
\end{aligned}
\tag{C.11}
$$

$\square$

### C.4. Proof of Proposition 4.4

*Proof.* We follow the proof in Barber et al. (2021) to show the result. We abbreviate the set of integers $\{n_0, \dots, n\}$ as $[n_0, n]$ in this subsection. Notice that under Assumptions 3.1 and 3.2, $(\check{X}_i^{n+1}, Y_i) \cup (\check{X}_{n+1}^i, Y_{n+1})$ are exchangeable. For $i, j \in [n_0, n+1]$ with $i \ne j$, denote $\check{X}_i^j = \mathbf{I}(X_i, \hat{\mathcal{O}}_i \cup \hat{\mathcal{O}}_j \cup \mathcal{O}^*)$. Then we assert that $\hat{\mu}(\check{X}_i^j) \ne \hat{\mu}(\check{X}_j^i)$ for $i \ne j$.

Define a matrix of residuals, $D \in \mathbb{R}^{(n+2-n_0) \times (n+2-n_0)}$, with entries

$$D_{pq} = \begin{cases} +\infty, & p = q \\ |Y_{p+n_0-1} - \hat{\mu}(\check{X}_{p+n_0-1}^{q+n_0-1})|, & p \neq q \end{cases} \tag{C.12}$$

for $p, q \in [n + 2 - n_0]$, i.e., the off-diagonal entries represent the residual for the $(p + n_0 - 1)$-th point which processed by **DI** with mask $\hat{\mathcal{O}}_{p+n_0-1} \cup \hat{\mathcal{O}}_{q+n_0-1} \cup \mathcal{O}^*$.

Define a comparison matrix, $A \in \{0, 1\}^{(n+2-n_0) \times (n+2-n_0)}$, with entries

$$A_{pq} = \mathbb{1}\{D_{pq} > D_{qp}\},$$

and $A_{pq} = 1$ means that the residual of $X_{p+n_0-1}$ is larger than that of $X_{q+n_0-1}$ which are processed with the same mask $\hat{\mathcal{O}}_{p+n_0-1} \cup \hat{\mathcal{O}}_{q+n_0-1} \cup \mathcal{O}^*$, and we say that data point $(p + n_0 - 1)$ "wins" $(q + n_0 - 1)$.

Define a set $S(A) \subseteq [n + 2 - n_0]$ of strange points as

$$S(A) = \left\{ p \in [n + 2 - n_0] : \sum_{q=1}^{n+2-n_0} A_{pq} \geq (1 - \alpha)(n + 2 - n_0) \right\}, \tag{C.13}$$

i.e., $p \in S(A)$ if it holds that, when we compare the residual $D_{pq}$ of the $(p + n_0 - 1)$-th point against residual $D_{qp}$ of the $(q + n_0 - 1)$-th point, the residual $D_{pq}$ is the larger one for a sufficiently high fraction of these comparisons. Note that each strange point $p \in S(A)$ can "lose" against at most $\alpha(n + 2 - n_0) - 1$ other strange points, this is because point $(p + n_0 - 1)$ must "win" against at least $(1 - \alpha)(n + 2 - n_0)$ points in total because it is strange, and point $(p + n_0 - 1)$ cannot win against itself as the definition of $D_{pq}$.

Denote by $s = |S(A)|$ the number of strange points. The key realization is now that, if we think about grouping each pair of strange points by the losing point, then we see that there are at most

$$s \cdot (\alpha(n + 2 - n_0) - 1)$$

pairs of strange points. So we have

$$\frac{s(s-1)}{2} \leq s \cdot (\alpha(n + 2 - n_0) - 1), \tag{C.14}$$

which simplifies to $s \leq 2\alpha(n + 2 - n_0) - 1 < 2\alpha(n + 2 - n_0)$.

Since the data points $\{(\check{X}_i^j, Y_i)\}_{i,j \in [n_0, n+1]}$ are exchangeable and the fitting algorithm is not affected by **DI**, it follows that $A \stackrel{d}{=} \Pi A \Pi^T$ for any $(n + 2 - n_0) \times (n + 2 - n_0)$ permutation matrix $\Pi$. In particular, for any index $q \in [n + 2 - n_0]$, suppose we take $\Pi$ to be any permutation matrix with $\Pi_{q,n+2-n_0} = 1$ (i.e., corresponding to a permutation mapping $n + 1$ to $q + n_0 - 1$), then we have

$$n + 2 - n_0 \in S(A) \iff q \in S(\Pi A \Pi^T),$$

and therefore

$$\mathbb{P}\{n + 2 - n_0 \in S(A)\} = \mathbb{P}\{q \in S(\Pi A \Pi^T)\} = \mathbb{P}\{q \in S(A)\}.$$

In other words, if we compare an arbitrary calibration point $q + n_0 - 1$ where $q \in [n + 2 - n_0]$ versus the test point $n + 1$, these two points are equally likely to be strange, by the exchangeability of the data. So we can calculate

$$\mathbb{P}\{n + 2 - n_0 \in S(A)\} = \frac{1}{n + 2 - n_0} \sum_{q=1}^{n+2-n_0} \mathbb{P}\{q \in S(A)\}$$

$$= \frac{\mathbb{E}\left[\sum_{q=1}^{n+2-n_0} \mathbb{1}\{q \in S(A)\}\right]}{n + 2 - n_0} = \frac{\mathbb{E}[|S(A)|]}{n + 2 - n_0} = \frac{s}{n + 2 - n_0} \leq 2\alpha, \tag{C.15}$$

according to (C.14). By the construction of $\hat{C}^{\text{JDI}}(\tilde{X}_{n+1})$ in (9), we have the following equivalence relation

$$Y_{n+1} \notin \hat{C}^{\text{JDI}}(\tilde{X}_{n+1}) \iff Y_{n+1} > \hat{q}_\alpha^+\left(\{\hat{\mu}(\check{X}_{n+1}^i) + \check{R}_i\}_{i=n_0}^n\right)$$

$$\text{or } Y_{n+1} < \hat{q}_\alpha^- \left( \{\hat{\mu}(\check{X}_{n+1}^i) - \check{R}_i\}_{i=n_0}^n \right). \tag{C.16}$$

In either case, we have

$$
\begin{aligned}
(1-\alpha)(n+2-n_0) &\leq \sum_{i=n_0}^n \mathbb{1}\{Y_{n+1} \notin \hat{\mu}(\check{X}_{n+1}^i) \pm \check{R}_i\} \\
&= \sum_{i=n_0}^n \mathbb{1}\{|Y_i - \hat{\mu}(\check{X}_i^{n+1})| < |Y_{n+1} - \hat{\mu}(\check{X}_{n+1}^i)|\} \\
&= \sum_{q=1}^{n+1-n_0} \mathbb{1}\{D_{q,n+2-n_0} < D_{n+2-n_0,q}\} = \sum_{q=1}^{n+1-n_0} A_{n+2-n_0,q},
\end{aligned} \tag{C.17}
$$

and therefore $n + 2 - n_0 \in S(A)$ according to (C.13). Combined with (C.15), we have

$$\mathbb{P}\{Y_{n+1} \notin \hat{C}^{\mathrm{JDI}}(\tilde{X}_{n+1})\} \leq \mathbb{P}\{n + 2 - n_0 \in S(A)\} \leq 2\alpha. \tag{C.18}$$

$\square$

## C.5. Proof of Theorem 5.1

C.5.1. PROOF OF PDI-CP

*Proof.* Under Assumption 3.1, denote

$$\check{X}_{n+1}^{\mathrm{DI}} = \mathbf{I}(X_{n+1}, \mathcal{O}^* \cup \tilde{\mathcal{T}}_{n+1}), \tag{C.19}$$

$$\check{X}_{n+1}^* = \mathbf{I}(X_{n+1}, \mathcal{O}^* \cup \hat{\mathcal{T}}_{n+1}). \tag{C.20}$$

The difference between $\check{X}_{n+1}^{\mathrm{DI}}$ and $\check{X}_{n+1}^*$ is

$$
\begin{aligned}
\check{X}_{n+1}^{\mathrm{DI}} - \check{X}_{n+1}^* &= (\tilde{\Delta}_{n+1}^{\mathcal{T}} \Delta_{n+1} - \hat{\Delta}_{n+1}^{\mathcal{T}} \Delta_{n+1}) X_{n+1} + (\hat{\Delta}_{n+1}^{\mathcal{T}} \Delta_{n+1} - \tilde{\Delta}_{n+1}^{\mathcal{T}} \Delta_{n+1}) \bar{\boldsymbol{x}}_{n+1} \\
&= \Delta_{n+1}(\tilde{\Delta}_{n+1}^{\mathcal{T}} - \hat{\Delta}_{n+1}^{\mathcal{T}})(X_{n+1} - \bar{\boldsymbol{x}}_{n+1}),
\end{aligned} \tag{C.21}
$$

where

$$
\tilde{\Delta}_{n+1}^{\mathcal{T}} = \mathrm{diag}(\{\tilde{\delta}_{n+1,j}^{\mathcal{T}}\}_{j\in[d]}) = \mathrm{diag}(\{\mathbb{1}\{j \notin \tilde{\mathcal{T}}_{n+1}\}\}_{j\in[d]}),
$$
$$
\hat{\Delta}_{n+1}^{\mathcal{T}} = \mathrm{diag}(\{\hat{\delta}_{n+1,j}^{\mathcal{T}}\}_{j\in[d]}) = \mathrm{diag}(\{\mathbb{1}\{j \notin \hat{\mathcal{T}}_{n+1}\}\}_{j\in[d]}),
$$

and $\bar{\boldsymbol{x}}_{n+1} = (\bar{x}_1, \ldots, \bar{x}_j, \ldots, \bar{x}_d)^\top$ is the Mean Imputation value of $\tilde{X}_{n+1}$.

Denote $\check{R}_{n+1} = |Y_{n+1} - \hat{\mu}(\check{X}_{n+1}^{\mathrm{DI}})|$, and $R_{n+1}^* = |Y_{n+1} - \hat{\mu}(\check{X}_{n+1}^*)|$ is residual computed by ODI feature $\check{X}_{n+1}^*$. Invoking $\ell_1$-sensitivity of $\hat{\mu}$, we have

$$
\begin{aligned}
|\check{R}_{n+1} - R_{n+1}^*| &\leq S_{\hat{\mu}} \cdot \|\check{X}_{n+1}^{\mathrm{DI}} - \check{X}_{n+1}^*\|_1 \\
&= S_{\hat{\mu}} \cdot \sum_{j=1}^d |\delta_{n+1,j}| \cdot |\tilde{\delta}_{n+1,j}^{\mathcal{T}} - \hat{\delta}_{n+1,j}^{\mathcal{T}}| \cdot |X_{n+1,j} - \bar{x}_j| \\
&= S_{\hat{\mu}} \cdot \sum_{j\in[d]\setminus\mathcal{O}^*} |\tilde{\delta}_{n+1,j}^{\mathcal{T}} - \hat{\delta}_{n+1,j}^{\mathcal{T}}| \cdot |X_{n+1,j} - \bar{x}_j| \\
&\leq S_{\hat{\mu}} \cdot |\tilde{\mathcal{T}}_{n+1} \triangle \hat{\mathcal{T}}_{n+1}| \cdot E_{n+1} \\
&:= \Delta_{n+1},
\end{aligned} \tag{C.22}
$$

where $E_{n+1} := \max_{j\in[d]} |X_{n+1,j} - \bar{x}_j|$ is the same as the definition in Theorem 4.3. Notice that $\check{X}_{n+1}^*$ is exchangeable with ODI features $\{\check{X}_i^*\}_{i=n_0}^n$ under Assumption 3.1, then the coverage gap of ODI-CP is

$$\left| \mathbb{P}\left\{ Y_{n+1} \in \hat{C}^{\mathrm{ODI}}(\tilde{X}_{n+1}) \right\} - (1-\alpha) \right|$$

$$
\begin{aligned}
&= \left| \mathbb{P}\left\{ \check{R}_{n+1} \le \hat{q}_\alpha^+(\{R_i^*\}_{i=n_0}^n) \right\} - (1-\alpha) \right| \\
&\le \left| \mathbb{P}\left\{ R_{n+1}^* - \Delta_{n+1} \le \hat{q}_\alpha^+(\{R_i^*\}_{i=n_0}^n) \right\} - \mathbb{P}\left\{ R_{n+1}^* \le \hat{q}_\alpha^+(\{R_i^*\}_{i=n_0}^n) \right\} \right| \\
&= F_{R^*}(\hat{q}_\alpha^+(\{R_i^*\}_{i=n_0}^n) + \Delta_{n+1}) - F_{R^*}(\hat{q}_\alpha^+(\{R_i^*\}_{i=n_0}^n)),
\end{aligned}
\tag{C.23}
$$

where $F_{R^*}$ is the distribution function of $\texttt{ODI-CP}$ residuals $\{R_i^*\}_{i=n_0}^{n+1}$. Similar to the proof of Theorem 4.3, we can obtain the coverage of $\texttt{PDI-CP}$:

$$
\begin{aligned}
\mathbb{P}\left\{ Y_{n+1} \in \hat{C}^{\mathrm{PDI}}(\tilde{X}_{n+1}) \right\} \ge{}& 1 - \alpha - \left[ F_{R^*}(\hat{q}_\alpha^+(\{R_i^*\}_{i=n_0}^n) + \Delta_{n+1}) - F_{R^*}(\hat{q}_\alpha^+(\{R_i^*\}_{i=n_0}^n)) \right] \\
&- \left[ F_{R^*}\left( \hat{q}_\alpha^+\left( \{R_i^* + S_{\hat{\mu}} \cdot E_i \cdot |\tilde{\mathcal{T}}_{n+1}|\}_{i=n_0}^n \right) \right) - F_{R^*}\left( \hat{q}_\alpha^+\left( \{R_i^* - S_{\hat{\mu}} \cdot E_i \cdot |\tilde{\mathcal{T}}_{n+1}|\}_{i=n_0}^n \right) \right) \right].
\end{aligned}
$$

$\square$

### C.5.2. PROOF OF $\texttt{JDI-CP}$

*Proof.* Under Assumption 3.1, denote

$$
\begin{aligned}
X_i^{n+1} &= \mathbf{I}(X_i, \hat{\mathcal{O}}_i \cup \mathcal{O}^* \cup \hat{\mathcal{T}}_{n+1}), \quad i = n_0, \dots, n, \tag{C.24} \\
X_{n+1}^i &= \mathbf{I}(X_{n+1}, \hat{\mathcal{O}}_i \cup \mathcal{O}^* \cup \hat{\mathcal{T}}_{n+1}), \quad i = n_0, \dots, n. \tag{C.25}
\end{aligned}
$$

According to Proposition 4.4, we have

$$
\mathbb{P}\left\{ Y_{n+1} \in \hat{C}^{\mathrm{JDI}}(X_{n+1}) \right\} \ge 1 - 2\alpha,
\tag{C.26}
$$

where

$$
\hat{C}^{\mathrm{JDI}}(X_{n+1}) = \left[ \hat{q}_\alpha^-(\{\hat{\mu}(X_{n+1}^i) - |Y_i - \hat{\mu}(X_i^{n+1})|\}_{i=n_0}^n), \ \hat{q}_\alpha^+(\{\hat{\mu}(X_{n+1}^i) + |Y_i - \hat{\mu}(X_i^{n+1})|\}_{i=n_0}^n) \right].
\tag{C.27}
$$

Note that if Assumption 3.2 is satisfied, $X_i^{n+1}$ and $X_{n+1}^i$ are equal to $\check{X}_i^{n+1}$ and $\check{X}_{n+1}^i$. Based on this, we can characterize the differences between features and obtain the coverage property of $\hat{C}^{\mathrm{JDI}}(\tilde{X}_{n+1})$ beyond Assumption 3.2. First, notice that the difference between $\check{X}_{n+1}^i$ and $X_{n+1}^i$ is

$$
\begin{aligned}
\check{X}_{n+1}^i - X_{n+1}^i &= \hat{\Delta}_i \Delta_{n+1} (\tilde{\Delta}_{n+1}^{\mathcal{T}} - \hat{\Delta}_{n+1}^{\mathcal{T}}) X_{n+1} + \hat{\Delta}_i \Delta_{n+1} (\hat{\Delta}_{n+1}^{\mathcal{T}} - \tilde{\Delta}_{n+1}^{\mathcal{T}}) \bar{\boldsymbol{x}}_{n+1} \\
&= \hat{\Delta}_i \Delta_{n+1} (\tilde{\Delta}_{n+1}^{\mathcal{T}} - \hat{\Delta}_{n+1}^{\mathcal{T}})(X_{n+1} - \bar{\boldsymbol{x}}_{n+1}),
\end{aligned}
\tag{C.28}
$$

and the difference between $\check{X}_i^{n+1}$ and $X_i^{n+1}$ is

$$
\begin{aligned}
\check{X}_i^{n+1} - X_i^{n+1} &= \hat{\Delta}_i \Delta_{n+1} (\tilde{\Delta}_{n+1}^{\mathcal{T}} - \hat{\Delta}_{n+1}^{\mathcal{T}}) X_i + \hat{\Delta}_i \Delta_{n+1} (\hat{\Delta}_{n+1}^{\mathcal{T}} - \tilde{\Delta}_{n+1}^{\mathcal{T}}) \bar{\boldsymbol{x}}_i \\
&= \hat{\Delta}_i \Delta_{n+1} (\tilde{\Delta}_{n+1}^{\mathcal{T}} - \hat{\Delta}_{n+1}^{\mathcal{T}})(X_i - \bar{\boldsymbol{x}}_i).
\end{aligned}
\tag{C.29}
$$

Invoking $\ell_1$-sensitivity of $\hat{\mu}$, we have

$$
\begin{aligned}
|\hat{\mu}(\check{X}_{n+1}^i) - \hat{\mu}(X_{n+1}^i)| &\le S_{\hat{\mu}} \cdot \left\| \check{X}_{n+1}^i - X_{n+1}^i \right\|_1 \\
&= S_{\hat{\mu}} \cdot \sum_{j=1}^d |\hat{\delta}_{i,j}| \cdot |\delta_{n+1,j}| \cdot |\tilde{\delta}_{n+1,j}^{\mathcal{T}} - \hat{\delta}_{n+1,j}^{\mathcal{T}}| \cdot |X_{n+1,j} - \bar{x}_j| \\
&= S_{\hat{\mu}} \cdot \sum_{j \in [d] \setminus \mathcal{O}^*} |\hat{\delta}_{i,j}| \cdot |\tilde{\delta}_{n+1,j}^{\mathcal{T}} - \hat{\delta}_{n+1,j}^{\mathcal{T}}| \cdot |X_{n+1,j} - \bar{x}_j| \\
&\le S_{\hat{\mu}} \cdot |\tilde{\mathcal{T}}_{n+1} \triangle \hat{\mathcal{T}}_{n+1}| \cdot E_{n+1},
\end{aligned}
\tag{C.30}
$$

and

$$
\left| |Y_i - \hat{\mu}(\check{X}_i^{n+1})| - |Y_i - \hat{\mu}(X_i^{n+1})| \right| \le S_{\hat{\mu}} \cdot |\tilde{\mathcal{T}}_{n+1} \triangle \hat{\mathcal{T}}_{n+1}| \cdot E_{n+1}.
\tag{C.31}
$$

Notice that the upper bound in (C.30) and (C.31) is exactly $\Delta_{n+1}$ in (C.22). Combining with (C.26), we can obtain the coverage gap:

$$\left| \mathbb{P}\left\{ Y_{n+1} \in \hat{C}^{\mathrm{JDI}}(\tilde{X}_{n+1}) \right\} - \mathbb{P}\left\{ Y_{n+1} \in \hat{C}^{\mathrm{JDI}}(X_{n+1}) \right\} \right|$$

$$= \left| \mathbb{P}\left\{ \hat{q}_\alpha^-(\{\hat{\mu}(\check{X}_{n+1}^i) - |Y_i - \hat{\mu}(\check{X}_i^{n+1})|\}_{i=n_0}^n) < Y_{n+1} \le \hat{q}_\alpha^+(\{\hat{\mu}(\check{X}_{n+1}^i) + |Y_i - \hat{\mu}(\check{X}_i^{n+1})|\}_{i=n_0}^n) \right\} \right.$$

$$\left. - \mathbb{P}\left\{ \hat{q}_\alpha^-(\{\hat{\mu}(X_{n+1}^i) - |Y_i - \hat{\mu}(X_i^{n+1})|\}_{i=n_0}^n) < Y_{n+1} \le \hat{q}_\alpha^+(\{\hat{\mu}(X_{n+1}^i) + |Y_i - \hat{\mu}(X_i^{n+1})|\}_{i=n_0}^n) \right\} \right|$$

$$\le \left| \mathbb{P}\{ \hat{q}_\alpha^-(\{\hat{\mu}(X_{n+1}^i) - \Delta_{n+1} - (|Y_i - \hat{\mu}(X_i^{n+1})| + \Delta_{n+1})\}_{i=n_0}^n) < Y_{n+1} \le \right.$$

$$\hat{q}_\alpha^+(\{\hat{\mu}(X_{n+1}^i) + \Delta_{n+1} + (|Y_i - \hat{\mu}(X_i^{n+1})| + \Delta_{n+1})\}_{i=n_0}^n) \}$$

$$\left. - \mathbb{P}\left\{ \hat{q}_\alpha^-(\{\hat{\mu}(X_{n+1}^i) - |Y_i - \hat{\mu}(X_i^{n+1})|\}_{i=n_0}^n) < Y_{n+1} \le \hat{q}_\alpha^+(\{\hat{\mu}(X_{n+1}^i) + |Y_i - \hat{\mu}(X_i^{n+1})|\}_{i=n_0}^n) \right\} \right|$$

$$= \left| \mathbb{P}\left\{ \hat{Q}_\alpha^- - 2\Delta_{n+1} < Y_{n+1} \le \hat{Q}_\alpha^+ + 2\Delta_{n+1} \right\} - \mathbb{P}\left\{ \hat{Q}_\alpha^- < Y_{n+1} \le \hat{Q}_\alpha^+ \right\} \right|$$

$$= F_Y(\hat{Q}_\alpha^+ + 2\Delta_{n+1}) - F_Y(\hat{Q}_\alpha^+) + F_Y(\hat{Q}_\alpha^-) - F_Y(\hat{Q}_\alpha^- - 2\Delta_{n+1}), \tag{C.32}$$

where

$$\hat{Q}_\alpha^- = \hat{q}_\alpha^-(\{\hat{\mu}(X_{n+1}^i) - |Y_i - \hat{\mu}(X_i^{n+1})|\}_{i=n_0}^n), \tag{C.33}$$

$$\hat{Q}_\alpha^+ = \hat{q}_\alpha^+(\{\hat{\mu}(X_{n+1}^i) + |Y_i - \hat{\mu}(X_i^{n+1})|\}_{i=n_0}^n). \tag{C.34}$$

$\square$

## D. DDC Method

The DDC method is the most widely used method to detect cellwise outliers. We display the process of DDC in Figure 13, and the details of the robust functions (highlighted in orange) can be found in Rousseeuw & Bossche (2018).

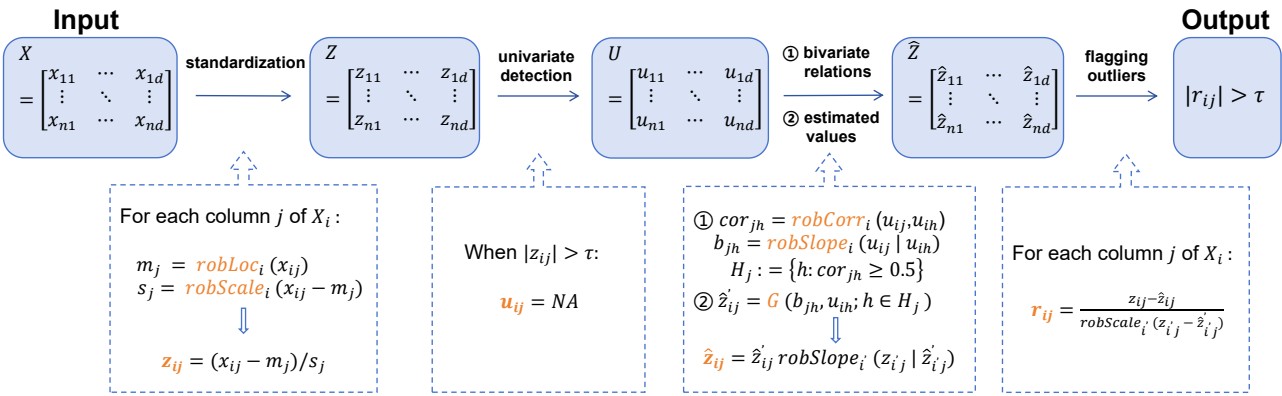

Figure 13. An implementation of DDC detection method.

## E. Additional experiment results

### E.1. Three benchmark methods in simulation

Now we present the specific construction forms of the three methods used for comparison in Section 6.

- Baseline: mask the entries of $\{X_i\}_{i=n_0}^n \cup \tilde{X}_{n+1}$ by $\mathcal{O}^*$, and obtain the processed features

$$\check{X}_i^{\mathrm{Base}} = \mathbf{I}(X_i, \mathcal{O}^*), \quad i = n_0, \dots, n,$$

$$\check{X}_{n+1}^{\mathrm{Base}} = \mathbf{I}(\tilde{X}_{n+1}, \mathcal{O}^*) = \mathbf{I}(X_{n+1}, \mathcal{O}^*),$$

then construct SCP interval

$$\hat{C}^{\text{Base}}(\tilde{X}_{n+1}) = \hat{\mu}(\check{X}_{n+1}^{\text{Base}}) \pm \hat{q}_{\alpha}^{+}(\{|Y_i - \hat{\mu}(\check{X}_i^{\text{Base}})|\}_{i=n_0}^{n}).$$

- SCP: directly compute residuals $R_i = |Y_i - \hat{\mu}(X_i)|$ on the calibration set, then construct SCP interval

$$\hat{C}^{\text{SCP}}(\tilde{X}_{n+1}) = \hat{\mu}(\tilde{X}_{n+1}) \pm \hat{q}_{\alpha}^{+}(\{R_i\}_{i=n_0}^{n}) \tag{E.1}$$

- WCP: consider the weights of residuals

$$p_i^w(x) = \frac{w(X_i)}{\sum_{l=n_0}^{n} w(X_l) + w(x)}, \quad i = n_0, \ldots, n,$$

$$p_{n+1}^w(x) = \frac{w(x)}{\sum_{l=n_0}^{n} w(X_l) + w(x)},$$

where the likelihood ratio $w(x) = dP_{\tilde{X}}(x)/dP_X(x)$ with $\tilde{X}_{n+1} \sim P_{\tilde{X}}$ can be estimated by Random Forests (more generally, any classifier that outputs estimated probabilities of class membership) as mentioned in Tibshirani et al. (2019), then construct WCP interval

$$\hat{C}^{\text{WCP}}(\tilde{X}_{n+1}) = \hat{\mu}(\tilde{X}_{n+1}) \pm \hat{q}_{\alpha}^{+}\left(\sum_{i=n_0}^{n} p_i^w(\tilde{X}_{n+1})\delta_{R_i} + p_{n+1}^w(\tilde{X}_{n+1})\delta_{\infty}\right). \tag{E.2}$$

## E.2. TPR and FDR for detection methods

In Table 3-5, we summarize the empirical TPR and FDR obtained through the application of the detection methods in the simulation.

*Table 3.* Empirical TPR and FDR of *DDC* under three settings when $\epsilon = \{0.2, 0.15, 0.1\}$, where the noise $Z_{n+1,j}$ is 10 .

| Setting | TPR | | | FDR | | |
|---|---|---|---|---|---|---|
| | 0.2 | 0.15 | 0.1 | 0.2 | 0.15 | 0.1 |
| A | 1 | 1 | 1 | 0.082 | 0.113 | 0.179 |
| B | 1 | 1 | 1 | 0.067 | 0.108 | 0.232 |
| C | 1 | 1 | 1 | 0.080 | 0.116 | 0.182 |

*Table 4.* Empirical TPR and FDR of *DDC* under three settings when $\epsilon = \{0.2, 0.15, 0.1\}$, where the noise $Z_{n+1,j} \overset{i.i.d.}{\sim} N(\mu, \sigma)$, $\mu, \sigma$ are randomly sampled from $U(0, 10)$.

| Setting | TPR | | | FDR | | |
|---|---|---|---|---|---|---|
| | 0.2 | 0.15 | 0.1 | 0.2 | 0.15 | 0.1 |
| A | 0.991 | 0.990 | 0.990 | 0.084 | 0.115 | 0.186 |
| B | 0.990 | 0.990 | 0.990 | 0.078 | 0.116 | 0.236 |
| C | 0.990 | 0.990 | 0.990 | 0.085 | 0.116 | 0.182 |

*Table 5.* Empirical TPR and FDR of *one-class SVM classifier* and *cellMCD estimate* methods under three settings when $\epsilon = 0.1$, where the noise $Z_{n+1,j} \overset{i.i.d.}{\sim} N(\mu, \sigma)$ and $\mu, \sigma$ are randomly sampled from $U(0, 10)$.

| Setting | TPR | | FDR | |
|---|---|---|---|---|
| | SVM | MCD | SVM | MCD |
| A | 0.988 | 0.990 | 0.026 | 0.173 |
| B | 0.988 | 0.989 | 0.023 | 0.172 |
| C | 0.988 | 0.989 | 0.027 | 0.175 |

### E.3. Supplementary simulation results for 6.3

In addition to Mean Imputation, we present the empirical coverage and length of `Baseline` and our method under different contamination probabilities when **I** is kNN or MICE in Figure 14 and 15, respectively. The parameters are the same as those in 6.3.

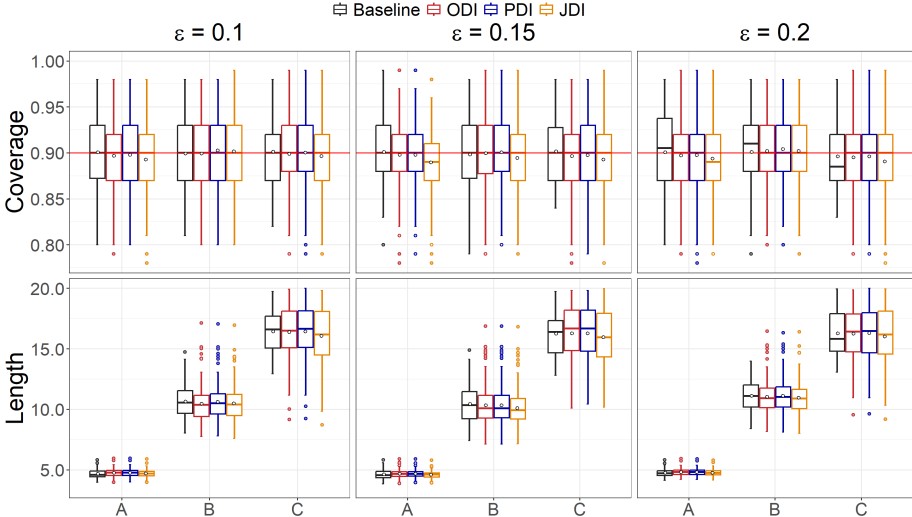

*Figure 14.* Simulation results of `Baseline` and our methods when $\epsilon \in \{0.1, 0.15, 0.2\}$. **D** is DDC and **I** is *kNN*.

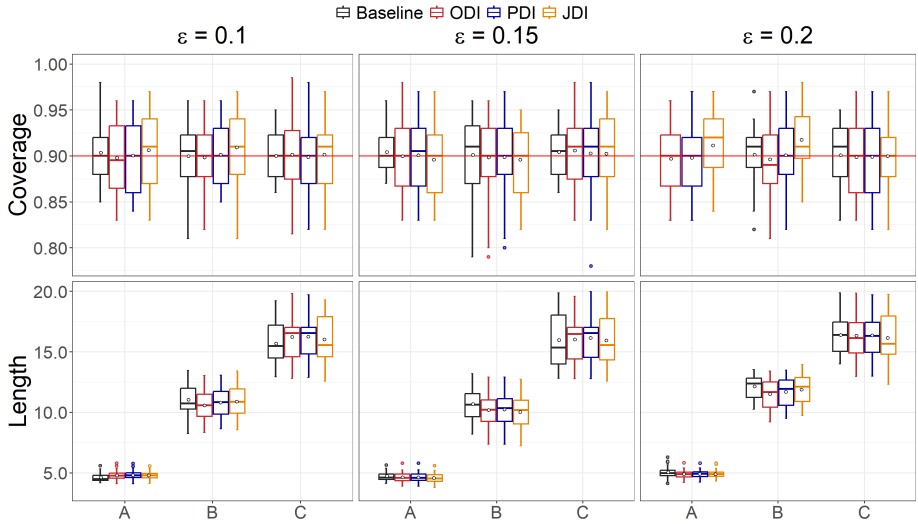

*Figure 15.* Simulation results of `Baseline` and our methods when $\epsilon \in \{0.1, 0.15, 0.2\}$. **D** is DDC and **I** is *MICE*.

### E.4. Supplement of airfoil dataset study

*Table 6.* Available variables in the airfoil dataset.

| Variable | Unit |
|---|---|
| Scaled sound pressure level of NASA airfoils (target) | dB |
| Frequency | Hz |
| Attack-angle | deg |
| Chord-length | m |
| Free-stream-velocity | m/s |
| Suction-side-displacement-thickness | m |

Creating training data, test data, and covariate shift: We repeated an experiment for 200 trials, and for each trial we randomly partition the data $\{(X_i, Y_i)\}_{i=1}^{1000}$ into two equally sized subsets $\mathcal{D}_t$ and $\mathcal{D}_c$, and construct a test set $\mathcal{D}_{test}$ containing cellwise outliers with the following steps.

- Log the first and fifth features of $X$ across all the data.

- According to the contamination probability $\epsilon = 0.02$, randomly select some entries to assign outlier 50.

- Use **DI** to handle covariates, where $\tau_j$ is set to $\sqrt{\chi^2_{1,0.95}}$ to control FDR.

- The set $\mathcal{O}^*$ used in `Baseline` and `ODI-CP` is the coordinates of those artificially added noises.

### E.5. Supplement of wind direction dataset study

*Table 7.* Available variables in the wind direction dataset.

| Variable | Unit |
|---|---|
| Wind direction (target) | rad |
| Cosine of wind direction in the previous hour | dimensionless |
| Sine of wind direction in the previous hour | dimensionless |
| Atmospheric pressure in the previous hour | mB |
| Air temperature (dry bulb) in the previous hour | °C |
| Dew point temperature in the previous hour | °C |
| Relative humidity in the previous hour | % |
| Wind gust in the previous hour | m/s |
| Wind speed in the previous hour | m/s |

Additional details are referred to E.4, but the test set $\mathcal{D}_{test}$ is constructed with the following steps.

- According to the contamination probability $\epsilon = 0.02$, randomly select some entries to assign outlier 100.

- Use **DI** to handle covariates, where $\tau_j$ is set to $\sqrt{\chi^2_{1,0.95}}$ to control FDR.

- The set $\mathcal{O}^*$ used in `Baseline` and `ODI-CP` is the coordinates of those artificially added noises.

### E.6. Supplement of riboflavin dataset study

*Table 8.* Available variables in the riboflavin dataset.

| Variable | Comments |
|---|---|
| Logarithm of riboflavin production rate | target |
| Log-transformed gene expression levels | $p = 4088$ (co)variables |

Additional details are referred to E.4 without introducing artificial outliers.

- Use **DI** to handle covariates, where $\tau_j$ is set to $\sqrt{\chi^2_{1,0.95}}$ to control FDR.

- The set $\mathcal{O}^*$ is unknown, so the experiments of `Baseline` and `ODI-CP` are not conducted.

- There may be cellwise outliers in $\mathcal{D}_t$ and $\mathcal{D}_c$.

