# OpenReview forum: "Conformal Prediction with Cellwise Outliers: A Detect-then-Impute Approach"
_ICML.cc/2025/Conference — ICML 2025 poster_

### Official Review · Reviewer_92Px · 2025-03-12

**Overall Recommendation:** 3

**Summary:**

Conformal Prediction (CP) provides prediction intervals with guaranteed coverage for black-box models under exchangeability assumptions. However, cellwise outliers isolated contaminated entries in test features break this exchangeability, leading to unreliable PI. This paper addresses this challenge by introducing a detect-then-Impute conformal prediction framework to robustly handle cellwise outliers.
This paper proposed two novel algorithms of conformal prediction, PDI-CP and JDI-CP, and provided a distribution-free coverage analysis under detection and imputation procedures. This paper establishes coverage error bounds for PDI-CP and JDI-CP achieves a finite sample 1 − 2α coverage guarantee. In section 6 and section 7, author experiments on synthetic and real datasets to demonstrate that proposed algorithms are robust and efficient.


=============

I thank the reviewer for the responses which clarified my concerns.
I have read them and keep my score unchanged.

**Claims And Evidence:**

yes

**Essential References Not Discussed:**

NA

**Experimental Designs Or Analyses:**

The experiment is reasonable overall, but a new experiment with setting linear, homoscedastic, Light-tailed could be added in simulation.

**Methods And Evaluation Criteria:**

yes

**Other Comments Or Suggestions:**

NA

**Other Strengths And Weaknesses:**

**Weakness**
1. **Computational Cost**: JDI-CP requires O(n) pairwise operations, which could be inefficient for large datasets.
2. **Logic**: The logic of the article introduction is confusing and should highlight the research issues of the article. Author should highlight the purpose of this paper is using DI method to solve cellwise outlier issues instead of giving the notation in introduction.
3. **Innovation**: This paper seems simply combining DI and CP, I don’t see any theoretical innovation. Author cites many theories which are proved in prior work like theorem 3.5 and focus too much on detection outlier instead of CP.
4. **Application**: Assumption 3.1 assume all the cellwise outliers are detected, but it sounds impossible in real world.

**Questions For Authors:**

NA

**Relation To Broader Scientific Literature:**

The key contributions of the paper are positioned at the intersection of cellwise outlier detection, conformal prediction, and missing data imputation, building on and extending prior work in these areas.

**Theoretical Claims:**

Theorems are correct under assumptions, but the assumptions are restrictive. Assumption 3.1 is critical for ODI-CP but unrealistic in practice. Assumption 3.2 may not hold for correlated features.

---

> ### Author Rebuttal · Authors · 2025-04-01
>
> >**Q1**: Author should highlight the purpose of this paper instead of giving the notation in introduction.
>
> We sincerely appreciate this constructive critique of our manuscript's organization. In response to your suggestion, we will comprehensively restructure the introduction in the future revision to:
>
> 1. **Problem Focus**
>    - Lead with the critical challenge of cellwise contamination in conformal prediction
>    - Highlight gaps in existing methods' ability to handle this data corruption
>
> 2. **Solution Framework**
>    - Clearly articulate our methodological innovations
>    - Emphasize how we address previously unsolved problems
>
> 3. **Presentation Strategy**
>    - Defer technical notation to methodology sections
>    - Maintain narrative flow while preserving rigor
>
> These adjustments will prioritize readability and better highlight the scientific significance of our approach.
> >**Q2**: Assumption 3.1 is unrealistic in practice.
>
> We appreciate your inquiry regarding Assumption 3.1 and we acknowledge its imperfection.
> - We'd like to stress that Assumption 3.1 is theoretically necessary for model-free and distribution-free coverage guarantees and can be satisfied by choosing a small detection threshold in practice.
> - Additionally, we also present a total variation coverage gap of our method when Assumption 3.1 is violated. If the total variation between processed calibration data and test data is small, the coverage can still be approximately controlled. Please refer to Reviewer **vdr2**'s Q1 for details and an additional experiment about Assumption 3.1 (due to the space limit).
>
> We would be happy to discuss this further if additional clarification would be helpful.
> >**Q3**: Assumption 3.2 may not hold for correlated features.
>
> Thanks for your question and we’d like to make the following explanations. For correlated features:
> - Assumption 3.2 can be satisfied if we use **one-class SVM** method to learn the score $s_j$ for each coordinate.
> - In simulations, we also used detection methods do not satisfy Assumption 3.2, such as **DDC** in Figure 3 and **cellMCD** in Figure 4(b).
> -  In Section 5, we provided theoretical results for our method when Assumption 3.2 does not hold.
>
> >**Q4**: A new experiment with setting linear, homoscedastic, light-tailed could be added.
>
> Thank you for your advice. In fact, we have included the experimental result of the setting you suggested in Appendix D.2 but didn't mention it in the main text due to the space limitation. We apologize for any possible reading incompleteness and will add this setting to the main text when we have an additional page in the revision.
> >**Q5**: JDI-CP requires O(n) pairwise operations, which could be inefficient for large datasets.
>
> We appreciate this insightful observation about our method's computational aspects.
> 1. **JDI-CP Design Considerations**:
>    - Primary objective: Finite-sample coverage guarantee
>    - Trade-off: Achieves robustness (demonstrated in Figure 7) at some computational cost
>    - Reference: Similar trade-offs exist in Barber et al. (2021)
> 2. **Future Directions**:
>    - Actively researching more efficient implementations
>    - Will optimize computational performance while maintaining theoretical guarantees
>
> [1]Barber R F, Candes E J, Ramdas A, et al. Predictive inference with the jackknife+. The Annals of Statistics, 2021.
> >**Q6**: This paper seems simply combining DI and CP. Author cites many theories which are proved in prior work like theorem 3.5. Focus too much on detection instead of CP.
>
> Thanks for your insightful comments on the contribution of our approach. We’d like to provide some clarifications to highlight the contribution of our work.
> - Firstly, our approach fundamentally differs from naive DI+CP combinations. As discussed and shown in Appendix A.1, such a direct combination cannot achieve coverage control. Therefore, we proposed new techniques to adaptively deploy DI on calibration data and proposed PDI-CP and JDI-CP.
> - Regarding Theorem 3.5, this is a negative result we obtained to illustrate the necessity of Assumption 3.1, and we have not found similar results in other papers. Additionally, we have provided a new counterexample under the CQR score, please refer to Reviewer **vdr2**'s Q2.
> - Finally, we believe that detection is essential to cope with cellwise outliers in predictive inference tasks. The classic CP methods under exchangeability assumption have been extensively studied. Here, we are concerned with the nonexchangeable problem caused by cellwise outliers, which is challenging and has not been studied before. The detection and imputation steps are used to identify and remove those outliers, so we can construct informative conformal prediction intervals. The detection is key to constructing exchangeable processed features in ODI-CP and JDI-CP, which rebuilds exchangeability between calibration data and test data and enables coverage control.
>
> Hope these explanations are acceptable for you!

---

### Official Review · Reviewer_xkXt · 2025-03-13

**Overall Recommendation:** 4

**Summary:**

This paper addresses conformal prediction with feature-wise outliers in the test sample. It assumes access to a detection oracle satisfying the sure detection and isolated detection assumption, and impute the values of the outlier features. After the detection and imputation procedure, split conformal prediction and Jackknife+ are applied, respectively, resulting in two kinds of prediction sets. A distribution-free finite sample coverage guarantee is proved for the latter. Beyond isolated detection, the coverage guarantee worsens with the difference of two detection set, which is empirically shown small. Experiments on synthetic and real datasets with synthetic perturbations show valid coverage and controllable sizes of the prediction set.

**Claims And Evidence:**

The claim lacks support in L48 which states weighted conformal prediction is unsuitable for CP with outliers because distribution shift cannot be estimated. For example, localized conformal prediction reweigh the calibration samples according to their distance to the test sample, without estimating the distribution shift. The WCP is clearly applicable, since the experiments also report Tibshirani et al.'s WCP method. There could be more discussion why weighted conformal prediction is less competitive in this setting.

**Essential References Not Discussed:**

Related works on weighted conformal prediction are listed but not fully addressed on why they are not applicable with outliers. Additionally, localized conformal prediction as a special form of WCP is not discussed. They are free of estimation of distribution shift which is claimed in the paper as a reason for incompetence.

[1] Leying Guan. Localized conformal prediction: A generalized inference framework for conformal prediction. Biometrika, 110(1):33–50, 2023.

[2] Rohan Hore and Rina Foygel Barber. Conformal prediction with local weights: randomization enables local guarantees. arXiv preprint arXiv:2310.07850, 2023.

**Experimental Designs Or Analyses:**

1. The experiments show that the proposed method is insensitive to different detection and imputation methods, which indicates the flexibility of the design.

2. Results with more recent weighted conformal prediction methods will consolidate the claim, such as localized conformal prediction.

3. Datasets with real outliers will be more convincing. The current results on real datasets introduce artificial perturbations.

**Methods And Evaluation Criteria:**

1. The major concern is over the strength of assumption 3.1 and 3.2. They assume oracle access to an outlier detector that has zero false negative rate. And assumption 3.2, which is necessary for finite length prediction sets, additionally excludes outliers in the joint space of features. Figure 2 has shown that the false discovery rate is around 0.4, which indicated that the assumptions are not approximately satisfied.

2. The form of non-conformity score is restricted to absolute deviation.

**Other Comments Or Suggestions:**

NA

**Other Strengths And Weaknesses:**

NA

**Questions For Authors:**

Are there contamination models other than Equation 1 that are worth consideration?

**Relation To Broader Scientific Literature:**

The most significant contribution of this paper is the first to consider conformal prediction with cellwise outliers, and proposes an algorithm with valid distribution-free coverage guarantee for finite samples (Theorem 4.4).

**Theoretical Claims:**

For theorem 4.3, the coverage lower bound can be vacuous for large calibration size since max E_i is non decreasing.

I have not checked the proofs.

---

> ### Author Rebuttal · Authors · 2025-04-01
>
> >**Q1**:Concerns focus on Assumptions 3.1 and 3.2, assuming detection with FNR=0. Assumption 3.2, crucial for finite length prediction interval, also excludes joint feature space outliers. Figure 2 shows FDR is around 0.4 , indicating unmet assumptions.
>
> Thanks for your valuable questions! We acknowledge the imperfection of Assumption 3.1 and explain as follows.
>
> - Assumption 3.1(FNR=0), also used in Liu et al.(2022) and Wasserman&Roeder(2009), can be met by choosing a small detection threshold in practice and is theoretically essential for model-free coverage guarantee. When Assumption 3.1 doesn't hold, we derived a new bound showing the coverage gap depends on the total variation between processed calibration and test data (see Reviewer **vdr2**’s Q1 due to space limit). We also added experiments by varying DDC detection thresholds $\sqrt{\chi_{1,p}^{2}}$ (adjusting $p$) based on Setting C in Appendix D.2. When FNR$\neq$0, our method can still approximately control the coverage.
> |$p$|0.5|0.7|0.9|0.99|
> |-|-|-|-|-|
> |FNR |0|0.005|0.008|0.013|
> |FDR|0.793|0.669|0.340|0.035|
> |PDI coverage|0.909|0.907|0.901|0.902|
> |JDI coverage|0.904|0.904|0.899|0.895|
>
> - We'd like to clarify the primary purpose of Assumption 3.2 is to ensure exchangeability of $\\{\hat{\cal{O}}\_i\cup\mathcal{O}^*\\}\_{i=n_0}^{n+1}$(see Lemma 3.3), not for finite length prediction intervals (PIs). In Section 5, we also provide theoretical coverage results when Assumption 3.2 doesn’t hold. In simulations, we also used detection methods that unsatisfy Assumption 3.2, such as **DDC** in Figure 3 and **cellMCD** in Figure 4(b), which don't lead to infinitely wide PIs.
> - Actually, Assumption 3.1 requires FNR equal to zero, while the frequency of $\tilde{\cal{T}}\_{n+1}=\hat{\cal{T}}\_{n+1}=\varnothing$ in Figure 2 means the false discovery (positive) rate (FDR) is zero.
>
> >**Q2**:Why weighted conformal prediction is less competitive?
>
> WCP requires precise likelihood ratio estimation, but arbitrary cellwise outliers in our setting render this impossible. Figures 3 and 11 highlight this limitation, showing WCP fails to meet target coverage.
> >**Q3**:Localized conformal prediction as a special form of WCP is not discussed.
>
> Thanks for your advice and we apologize for overlooking these references.
> - LCP controls conditional coverage under i.i.d. test and calibration data, differing from our focus on distribution shift caused by cellwise outliers in test data, which LCP cannot cope with. We’ll add a detailed discussion of LCP and these references in the revision.
> - We also compare our method with baseLCP, calLCP (Guan,2023), and RLCP (Hore&Barber, 2023) through experiments. Clean data follows Section 5.1.1 of Hore&Barber(2023), and cellwise outliers are generated via Equation (10) in our paper with $\epsilon=0.03$. Results show LCP methods fail to provide informative PIs.
> ||Bandwidth|0.1|0.2|0.4|0.8|1.6|
> |-|-|-|-|-|-|-|
> |baseLCP|Coverage|0.90|0.90|0.90|0.90|0.95|
> ||Infinite PI%|0.90|0.90|0.90|0.90|0.81|
> |calLCP|Coverage|0.94|0.94|0.94|0.94|0.93|
> ||Infinite PI%|0.88|0.88|0.88|0.87|0.66|
> |RLCP|Coverage|0.90|0.90|0.90|0.90|0.90|
> ||Infinite PI%|0.90|0.90|0.90|0.86|0.48|
> |PDI|Coverage|0.90|||||
> || Infinite PI\%|0|||||
> |JDI|Coverage|0.89|||||
> ||Infinite PI\%|0|||||
>
> >**Q4**:The form of non-conformity score is restricted to absolute deviation.
>
> For simplicity, we use the absolute residual score, but our method supports various non-conformity scores like CQR score. Please refer to Reviewer **vdr2**'s Q2 for a discussion about CQR.
> >**Q5**:Theorem 4.3's coverage lower bound may be vacuous with large calibration size.
>
> After checking our proof, we found the expansion to $\max_{i= n_0,\ldots,n}E_i$ is unnecessary and the last inequality in (B.10) (Appendix B.3) should be removed. The modified coverage gap is given by:$$\mathbb{P}\left\\{\hat{q}\_{\alpha}^+(\\{R_i^*-S_{\hat{\mu}}\cdot E_i\cdot|\tilde{\cal{O}}\_{n+1}\setminus\mathcal{O}^*|\\}\_{i=n_0}^n)<R_{n+1}^*\leq\hat{q}\_{\alpha}^+(\\{R_i^*+S_{\hat{\mu}}\cdot E_i\cdot|\tilde{\cal{O}}\_{n+1}\setminus\cal{O}^*|\\}\_{i=n_0}^n)\right\\},$$which will not be vacuous for large calibration size.
>
> Thank you for catching this key technical nuance!
> >**Q6**:Dataset with real outlier will be more convincing.
>
> To further demonstrate robustness, we test our method on a riboflavin gene expression dataset(Liu et al.,2022) with confirmed cellwise outliers. Our method maintain coverage above the target $1-\alpha=0.9$ while LCP methods ($h=0.1$) fail to provide meaningful PIs.
> ||SCP|WCP|baseLCP|calLCP|RLCP|PDI|JDI|
> |-|-|-|-|-|-|-|-|
> |Coverage|0.83|0.85|0.90|0.96|0.90|0.93|0.95|
> |Length|1.82|Inf|Inf|Inf|Inf|3.08|3.29|
>
> >**Q7**:Are other contamination models worth considering?
>
> The Tukey-Huber Contamination Model generates casewise outliers, assuming most samples follow a target distribution $F$, while the others follow an arbitrary distribution $H$:$$X\sim(1-\epsilon)F+\epsilon H,$$where $\epsilon\in[0,1)$ is the contamination ratio.

---

> > ### Comment · Reviewer_xkXt · 2025-04-02
> >
> > Thank the author for their response. Most of my concerns and questions are addressed. I have raised the rating from 3 to 4.

---

> > > ### Author Response · Authors · 2025-04-03
> > >
> > > Many thanks for the review and raising your rating! If you have any other questions, concerns, and comments, please let us know. We would like to provide our responses and address them in the future revision. Thank You!

---

### Official Review · Reviewer_8B4L · 2025-03-13

**Overall Recommendation:** 3

**Summary:**

This paper proposes a DI-CP framework to handle cellwise outliers in conformal prediction. The key idea is first to detect outliers in the test feature vector and then impute them before applying conformal prediction. To maintain exchangeability, a similar detection-imputation process is used to calibration samples.
The authors propose two methods: 1, PDI-CP (Proxy Detection-Imputation CP), which applies detection and imputation separately. 2, JDI-CP (Joint Detection-Imputation CP), which modifies detection rules to ensure theoretical coverage guarantees.
The paper provides theoretical guarantees, including a finite-sample coverage bound for JDI-CP and empirical results showing that DI-CP performs robustly under contamination.

**Claims And Evidence:**

This paper presents a novel and relevant problem in conformal prediction, but the theoretical guarantees depend on overly strong assumptions about detection accuracy. The method may fail if detection is imperfect or if too many inliers are misclassified as outliers.

**Essential References Not Discussed:**

I think authors have achieved a fairly complete literature review.

**Experimental Designs Or Analyses:**

Empirical evaluation lacks robustness analysis.
The paper does not study sensitivity to different detection thresholds and imputation choices.

**Methods And Evaluation Criteria:**

There exist two problems:

1. Strong assumptions on detection accuracy (Assumption 3.1 is unrealistic).
The method assumes perfect outlier detection. This is impractical because most real-world detection methods have false negatives.
If detection is imperfect, theoretical guarantees do not hold, making the results less applicable to real-world data.

2. Excessive false positives break the method.
The authors claim that choosing a larger detection threshold guarantees Assumption 3.1. However, this results in too many false positives.
Excessive imputation may change the calibration distribution, violating exchangeability assumptions (Lemma 3.3 fails).
There is no analysis of how misclassified inliers affect conformal validity.

3. Appendix A.1 considers Direct-ODI and Direct-PDI, which do not modify calibration samples. This setting contradicts the main method’s justification that calibration samples must be processed to maintain exchangeability.

**Other Comments Or Suggestions:**

There are a lot of math notations proposed in this paper. Could the author provide the math notation table in the appendix?

**Other Strengths And Weaknesses:**

Strength:

Novel problem setting: The problem of cellwise outliers in conformal prediction is important and underexplored. The paper provides a structured approach to address it.

Exchangeability considerations: The idea of applying detection and imputation jointly to calibration and test samples is a novel extension that ensures that conformal prediction remains valid despite outliers.

**Questions For Authors:**

1. The method assumes perfect outlier detection. How does coverage degrade when detection is imperfect?

2. If choosing a larger detection threshold ensures Assumption 3.1, doesn’t this break exchangeability by misclassifying inliers as outliers? How does this affect empirical coverage?

3. Theorem 4.3 and Theorem 4.4 assume that imputation does not change the residual distribution. How do results hold when imputations introduce bias?

4.  How does the method perform when detection is imperfect?

5. Have you tested other imputation strategies?

6. Can you provide a sensitivity study showing how conformal coverage varies under different imputation and detection methods?

**Relation To Broader Scientific Literature:**

This paper reviews the literature of cellwise outliers, conformal prediction without exchangeability, predictive inference with missing data, and conformal inference for outlier detection. It also lists the key difference of settings or tasks for works in the related area.

**Theoretical Claims:**

1. Theoretical results assume that imputation does not introduce significant bias. However, imputation methods systematically shift feature distributions, leading to biased conformity scores.

---

> ### Author Rebuttal · Authors · 2025-04-01
>
> >**Q1**: Assumption 3.1 is impractical. How does coverage degrade when detection is imperfect?
>
> Thank you for your insightful question!
>
> - We acknowledge the imperfection of Assumption 3.1 and have obtained a new coverage gap bound in total variation: if there are still outliers in test point $\tilde{X}\_{n+1}$ after the detection procedure, Lemma 3.3 will not hold because the exchangeability between $(\check{X}\_{n+1}^{\rm{DI}},Y_{n+1})$ and $\\{(\check{X}\_i^*,Y_{i})\\}\_{i=n_0}^n$ is broken, where $\\{\check{X}\_i^*\\}\_{i=n_0}^n$ are the ODI features. At this point, there is a total variation coverage gap of the prediction interval (PI):$$\mathbb{P}\\{Y_{n+1}\in\hat{C}^{\rm{ODI}}(\tilde{X}\_{n+1})\\}=\mathbb{P}\left\\{|\hat{\mu}(\check{X}\_{n+1}^{\rm{DI}})-Y_{n+1}|\le\hat{q}\_{\alpha}^+(\\{R_i^*\\}\_{i=n_0}^n)\right\\}\ge 1-\alpha-\frac{1}{n-n_0+1}\sum_{i=n_0}^n d_{\rm{TV}}(\check{X}\_{n+1}^{\rm{DI}},\check{X}\_i^*).$$Notice that, we can still have a approximate coverage control if $d_{\mathrm{TV}}(\check{X}_{n+1}^{\mathrm{DI}}, \check{X}_i^*)$ is small. A similar bound can be obtained for PDI-CP and JDI-CP and we will add these results in the revision.
> - In addition, we add a new experiment to show the influence of imperfect detection. Please see **Q6** for details.
>
> >**Q2**: If choosing a larger detection threshold ensures Assumption 3.1, doesn’t this break exchangeability? How does this affect empirical coverage?
>
> - According to the construction, we know that false positives do not break the data exchangeability in ODI-CP and JDI-CP. Since we use $\tilde{\cal{O}}\_{n+1}$ to approximate $\cal{O}^*$ in PDI-CP, it will break the exchangeability between processed test data and calibration data because $\tilde{\cal{O}}\_{n+1}$ depends only on test data. As we stated in Theorem 4.3, the coverage gap is affected by the number of false discoveries $|\tilde{\cal{O}}\_{n+1}\setminus\cal{O}^*|$.
> - In Appendix D.3, we summarized the empirical false discovery rate (FDR) and true positive rate (TPR) of the detection methods in our simulation. We will include this discussion in future revision, using the table from **Q6** as an illustrative example. As the threshold decreases, the number of discoveries increases, leading to a higher FDR. Notably, the table demonstrates the empirical coverage of our method remains robust to variations in FDR.
>
> >**Q3**: Theorem 4.3 and 4.4 assume imputation doesn't change the residual distribution.
>
> We apologize for any confusion. Here, $F_{R}$ denotes the distribution function of the ODI-CP residuals $\\{R_i^*\\}\_{i=n_0}^n$, **not** the residuals from the raw calibration data before the DI procedure. Thus, our method doesn't assume "imputation preserves the residual distribution." We appreciate your attention and will clarify this notation in the revision.
> >**Q4**: Have you tested other imputation strategies? How coverage varies under different imputation and detection methods?
>
> - In addition to Mean Imputation, we evaluated two other imputation methods in Section 6.2: **kNN** and **MICE**. Figure 5 shows our method maintains robust empirical coverage across all imputation strategies.
> - Our analysis in Sections 6.1-6.2 compares coverage and length of PI across various detection and imputation methods. Figures 4-5 show our method consistently maintains robust coverage control across all configurations with stable interval lengths.
>
> >**Q5**: Direct-ODI and Direct-PDI contradict the main method’s justification.
>
> We apologize for any ambiguity and would like to clarify:
> - Direct-ODI and Direct-PDI in Appendix A.1 are naive combinations of DI with CP, which are **not** our proposed methods.
> - Figure 8 shows Direct-PDI fails to maintain proper coverage while our PDI-CP successfully achieves target coverage. This empirical evidence confirms simply combining DI with CP (without our proposed modifications) cannot guarantee valid coverage.
>
> We appreciate your careful review and will revise Appendix A.1 for clarity.
> >**Q6**: The paper does not study sensitivity to different detection thresholds and imputation choices.
>
> Regarding the sensitivity of our method:
> 1. **Imputation methods**: Figure 5 in Section 6.2 demonstrates our method maintains robust performance across different imputation choices.
> 2. **Detection thresholds**: We conduct additional experiments by varying the DDC detection threshold $\sqrt{\chi_{1,p}^2}$ (adjusting $p$) based on Setting C in Appendix D.2. As shown in the table, our method demonstrates robust coverage performance when threshold changes.
> |$p$|0.5|0.7|0.9|0.99|
> |-|-|-|-|-|
> |FNR |0|0.005|0.008|0.013|
> |FDR|0.793|0.669|0.340|0.035|
> |PDI coverage|0.909|0.907|0.901|0.902|
> |JDI coverage|0.904|0.904|0.899|0.895|
>
> Please let us know if you would like any additional details.
> >**Q7**: Could you provide a math notation table?
>
> Thanks for your helpful advice. We will add a math notation table to the appendix of the revision.
>
> Hope these explanations are acceptable for you!

---

> > ### Comment · Reviewer_8B4L · 2025-04-04
> >
> > Thank the author for their response. Most of my concerns are addressed. I have raised my score from 2 to 3.

---

> > > ### Author Response · Authors · 2025-04-04
> > >
> > > We greatly appreciate your efforts in reviewing our work and raising your score! If there are any additional insights or suggestions you would like to share, we are eager to hear them. Thank you once again for your support!

---

### Official Review · Reviewer_vdr2 · 2025-03-14

**Overall Recommendation:** 3

**Summary:**

When some entries of the test features are contaminated, the paper introduces a detect-then-impute conformal prediction framework. This framework first applies an outlier detection procedure to identify contaminated entries in the test features and then uses an imputation method to fill in the identified outliers. Moreover, the authors apply the detection and imputation procedures to the calibration set, ensuring the construction of exchangeable features for the conformal prediction interval of the test label. Two practical algorithms including PDI-CP and JDI-CP are proposed, with lower bounds on marginal coverage probability established under certain conditions. Numerical experiments on both synthetic and real datasets are provided to demonstrate the performance of the proposed algorithms.

**Claims And Evidence:**

Yes.

**Essential References Not Discussed:**

No critical references appear missing.

**Experimental Designs Or Analyses:**

I have reviewed all experimental parts in the paper.

**Methods And Evaluation Criteria:**

The paper evaluates the proposed method on synthetic data and two real datasets and compares its performance against several relevant baseline methods.

**Other Comments Or Suggestions:**

No.

**Other Strengths And Weaknesses:**

No.

**Questions For Authors:**

1. The validity of Assumption 3.1 appears to depend heavily on the quality of the detection rule, which may be restrictive in practice. In Theorem 3.5, the negative result is derived using a specific form of the prediction set. Would the conclusion change if other adaptive prediction sets, such as those based on conformalized quantile regression (CQR), were used instead?

2. How does the contamination rate in the test features affect the length of the prediction sets?

**Relation To Broader Scientific Literature:**

The paper proposes a framework to construct prediction sets with marginal coverage guarantees when the test input feature is contaminated.

**Theoretical Claims:**

I checked the theoretical proofs of Lemma 3.3 and Proposition 3.4 and believe they are correct.

---

> ### Author Rebuttal · Authors · 2025-04-01
>
> >**Q1**: The validity of Assumption 3.1 (sure detection) depends on quality of detection, which may be restrictive in practice.
>
> Thanks for your valuable question! We acknowledge the imperfection of Assumption 3.1 and make the following explanations.
> - Sure detection/screening conditions are commonly used in existing works (Wasserman&Roeder,2009; Liu et al.,2022) to ensure all relevant variables are retained. We adopt a similar condition in Assumption 3.1 for model-free coverage guarantee.
> - Assumption 3.1 is essential for meaningful prediction intervals (PIs): if outliers persist in $\check{X}\_{n+1}^{\rm{DI}}$, Theorem 3.5 shows PIs can become infinitely wide in expectation.
> - Violating Assumption 3.1 impacts our method by introducing a total variation coverage gap: if outliers persist in the test point $\tilde{X}\_{n+1}$ after detection, Lemma 3.3 fails due to the exchangeability between $(\check{X}\_{n+1}^{\rm{DI}},Y_{n+1})$ and $\\{(\check{X}\_i^*,Y_{i})\\}\_{i=n_0}^n$ is broken, where $\\{\check{X}\_i^*\\}\_{i=n_0}^n$ are the ODI features. At this point, there is a total variation coverage gap of the PI:$$\mathbb{P}\\{Y_{n+1}\in\hat{C}^{\rm{ODI}}(\tilde{X}\_{n+1})\\}=\mathbb{P}\left\\{|\hat{\mu}(\check{X}\_{n+1}^{\rm{DI}})-Y_{n+1}|\le\hat{q}\_{\alpha}^+(\\{R_i^*\\}\_{i=n_0}^n)\right\\}\ge 1-\alpha-\frac{1}{n-n_0+1}\sum_{i=n_0}^n d_{\rm{TV}}(\check{X}\_{n+1}^{\rm{DI}},\check{X}\_i^*).$$The impact is mild if $d_{\rm{TV}}(\check{X}_{n+1}^{\rm{DI}}, \check{X}_i^*)$ is small; similar bounds apply to PDI-CP and JDI-CP.
> - In practice, Assumption 3.1 (TPR=1) can be satisfied when the detection threshold is small; our method still maintains target coverage in simulations and real examples even if some outliers were not completely detected. For your convenience, we also added a new experiment under different DDC detection thresholds $\sqrt{\chi\_{1,p}^{2}}$ (varying $p$) based on Setting C in Appendix D.2, which shows Assumption 3.1 is exactly satisfied (TPR=1) at $p=0.5$. When TPR<1, our method can still achieve approximate control of coverage.
> |$p$|0.5|0.7|0.9|0.99|
> |-|-|-|-|-|
> |TPR|1|0.995|0.992|0.987|
> |FDR|0.793|0.669|0.340|0.035|
> |PDI coverage|0.909|0.907|0.901|0.902|
> |JDI coverage|0.904|0.904|0.899|0.895|
>
> [1] Wasserman, L. and Roeder, K. High dimensional variable selection. The Annals of Statistics, 2009.
>
> [2] Liu, Y., Ren, H., Guo, X., Zhou, Q., and Zou, C. Cellwise outlier detection with false discovery rate control. Canadian Journal of Statistics, 2022.
> >**Q2**: Would the conclusion in Theorem 3.5 hold with other adaptive prediction sets like conformalized quantile regression?
>
> Thank you for your question! We prove a similar result for the CQR score, and add new simulation results based on CQR.
> - The PI constructed from CQR is $\hat{C}(X)=[\hat{f}^{lo}(X)-\hat{q}\_n,\hat{f}^{up}(X)+\hat{q}\_n]$, where $\hat{f}^{lo}$ and $\hat{f}^{up}$ are the lower and upper quantile regression models, and $\hat{q}\_n$ is the quantile of empirical distribution of CQR computed on the calibration set.
> - Following the proof of Theorem 3.5 in Appendix B.2, $Y_i=X_{i,1}+X_{i,2}$ where $X_{i,1},X_{i,2}\sim\rm{Uniform}([0,1])$ for $i\in[n+1]$. Suppose $\hat{f}^{lo}(x)=\beta_1^{lo}x_1+\beta_2^{lo}x_2$ where $\beta_2^{lo}\neq0$, and the test point $\tilde{X}\_{n+1}=(X_{n+1,1},Z_{n+1,2})^{\top}$ where$$Z_{n+1,2}=\frac{M+1}{\beta_2^{lo}} \mathbb{1}\\{\beta_1^{lo}\geq 1\\}+\frac{M+2}{\beta_2^{lo}}\mathbb{1}\\{0<\beta_1^{lo}<1\\}+\frac{M-\beta_1^{lo}+2}{\beta_2^{lo}}\mathbb{1}\\{\beta_1^{lo}\leq 0\\}$$for some large positive value $M$. If $\check{X}\_{n+1}^{\rm{DI}}$ still contains $Z_{n+1,2}$ and $\hat{C}(\tilde{X}\_{n+1})$ covers the true label, we have$$\max\\{\hat{f}^{lo}(\check{X}\_{n+1}^{\rm{DI}})-Y_{n+1},Y_{n+1}-\hat{f}^{up}(\check{X}\_{n+1}^{\rm{DI}})\\}\geq\hat{f}^{lo}(\check{X}\_{n+1}^{\rm{DI}})-Y_{n+1}\geq M,$$which means $\mathbb{P}(\hat{q}\_n\geq M)\geq\mathbb{P}(Y_{n+1}\in\hat{C}(\tilde{X}\_{n+1}))\geq1-\alpha$.
> - We also conduct an experiment using CQR to construct PIs, where the Baseline used in our simulation can be considered as the optimal method for constructing split conformal PI for cellwise outlier, which masks calibration and test features by $\cal{O}^*$. This experiment will be added in future revisions.
> ||Baseline|ODI|PDI|JDI|
> |-|-|-|-|-|
> |Coverage| 0.905|0.902|0.900|0.885|
> |Length|4.366|4.291|4.282|5.544|
>
> >**Q3**: How does the contamination rate affect the length of the prediction sets?
>
> - Figure 6 in Section 6.3 shows the coverage and length across cell contamination probabilities with DDC and Mean Imputation, while Figures 12-13 in Appendix D.4 display results for kNN and MICE imputation.
> - Results indicate the length of our method remains stable across varying contamination rates, whereas that of Baseline increases with higher contamination. Figure 3 demonstrates our method's competitive length and target coverage, surpassing the classical WCP method.
>
> Hope these explanations can ease your doubts!

---

### Decision · Program_Chairs · 2025-05-01

**Decision:**

Accept (poster)

**Comment:**

This paper studies a scenario in conformal prediction when some entries of the test features are contaminated, which leads to a violation of the exchangeability assumption that is critical for the correct coverage probability. The proposed approach is first to apply an outlier detection method to identify contaminated entries in the test features and then impute these outliers before applying conformal prediction.

Two practical algorithms are proposed, with lower bounds on marginal coverage probability established under certain conditions. Numerical experiments on both synthetic and real datasets are provided to demonstrate the performance of the proposed algorithms. A bonus is that the same approach can also be applied to the calibration set. Since the calibration data presumably came from the same source as the training data, it would seem that this approach needs to be extendable to the setting when the training set contains outliers, or realistically, a worse scenario when both the training and calibration data are contaminated. It is important that the authors address this question in the final version.

The problem is highly relevant but one reviewer thinks the theoretical guarantees depend on overly strong assumptions about detection accuracy. The method may fail if detection is imperfect or if too many inliers are misclassified as outliers.